# Functional genetic variants can mediate their regulatory effects through alteration of transcription factor binding

Andrew D. Johnston [1], Claudia A. Simões-Pires [1], Taylor V. Thompson [1], Masako Suzuki [1] & John M. Greally [1]

Functional variants in the genome are usually identified by their association with local gene expression, DNA methylation or chromatin states. DNA sequence motif analysis and chromatin immunoprecipitation studies have provided indirect support for the hypothesis that functional variants alter transcription factor binding to exert their effects. In this study, we provide direct evidence that functional variants can alter transcription factor binding. We identify a multifunctional variant within the *TBC1D4* gene encoding a canonical NFκB binding site, and edited it using CRISPR-Cas9 to remove this site. We show that this editing reduces *TBC1D4* expression, local chromatin accessibility and binding of the p65 component of NFκB. We then used CRISPR without genomic editing to guide p65 back to the edited locus, demonstrating that this re-targeting, occurring ~182 kb from the gene promoter, is enough to restore the function of the locus, supporting the central role of transcription factors mediating the effects of functional variants.

---

[1] Center for Epigenomics and Department of Genetics (Division of Genomics), Albert Einstein College of Medicine, 1301 Morris Park Avenue, Bronx, NY 10461, USA. Correspondence and requests for materials should be addressed to J.M.G. (email: john.greally@einstein.yu.edu)

The variability in genomic sequences between individuals that is explored for disease associations includes loci that have obvious functional properties. Some of these loci change the coding sequences of genes, but a large proportion exist in the non-coding majority of the genome and are associated with changes in genomic function. The main functional property of the genome that has been explored for association with DNA sequence polymorphism is gene expression. Differences in the level of expression of a gene that depends on the allele present at a locus nearby defines that nearby locus as an expression quantitative trait locus, or eQTL[1]. A locus influencing DNA methylation *in cis* is referred to as a methylation QTL (meQTL)[2–5], while there are also examples of loci influencing *cis* chromatin states[6].

When studied at the sequence level, these functional variants can be described to contain binding sites for sequence-specific transcription factors (TFs), with the presence of a sequence variant within that site assumed to alter the binding of the TF at this locus[7–10]. By changing TF binding, it is furthermore assumed that gene expression, DNA methylation, and chromatin states are altered as consequences[11,12]. Supporting the model of DNA sequence polymorphism influencing TF binding as the primary mechanism for these downstream functional consequences, chromatin immunoprecipitation has shown differences between individuals[12,13] or alleles[14,15] at a locus for TF binding associated with sequence variation at that site. While this is strong evidence for the TF binding model, it remains an association and not a direct test of the hypothesis. As an alternative hypothesis, it is proposed that genetic polymorphism at functional genetic variants influences enhancer RNA structure, altering its ability to act as an aptamer to recruit proteins locally, preventing it from facilitating local looping of chromatin[16].

These observations contrast with the assumptions being made in a second current area of genomic research, the use of 'epigenetic editing' to modify regulatory events at specific loci in the genome[17,18]. These approaches use reagents that have sequence-specificity to target a pre-defined locus, including zinc finger domain proteins[19], the TALE binding domain of transcription activator-like effector nucleases (TALENs)[20–22], and, more recently, the CRISPR system with an inactivated nuclease[17,18]. With the idea that 'epigenetic' regulators are those influencing chromatin states and DNA methylation, a number of enzymes that modify these biochemical properties at a locus have been targeted, with varying efficacies and gene regulatory outcomes[23–39].

There is, however, some lack of coherence between the ideas being pursued by those studying functional variants and those performing epigenetic editing. The former group would regard TFs as the primary mediators of events that include transcriptional, DNA methylation, and chromatin state changes as consequences. The premise of many 'epigenetic editing' approaches is that altering chromatin states and DNA methylation is sufficient to alter transcriptional regulation, ignoring any need for TFs to function locally. It remains to be seen whether the successes or failures of these two independent approaches will reveal insights into physiological transcriptional regulation, for example, whether chromatin modifications by themselves can allow transcription to occur in the absence of local TF binding.

Here, we seek to test the model that TF binding mediates functional genomic outcomes at a polymorphic and regulatory locus that we identified in the human genome. We use CRISPR for targeted mutagenesis at a TF binding site overlapping this multifunctional variant, followed by the further use of CRISPR as a targeting system to deliver to the edited locus the TF that could no longer bind to the site—a form of epigenetic editing that assumes a primary role for the TF in regulating the locus. The results reveal insights into both the normal physiology of functional genomic variants as well as guidance for epigenetic editing strategies.

## Results

**Identification of multifunctional variants in LCLs.** Our primary goal was to identify a small set of polymorphic loci with very strong functional effects on gene expression, chromatin accessibility, and DNA methylation. We performed our studies on the CEPH/Utah Pedigree 1463, which includes 17 members in three generations (Supplementary Fig. 1)[40], on which deep (Illumina Platinum genomes) sequencing has been performed[41], and for which lymphoblastoid cell lines (LCLs) are available from the Coriell Institute. By studying a family, we could identify effects mediated by variants that are rare in wider populations.

To reduce technical variability, substantial care was taken with the assays based on cultured LCLs. A single batch of reagents was used for all cell cultures, and all genome-wide assays were performed on the same flask of cells (except for a further biological replicate used for our ATAC-seq[42] studies). When samples were harvested as a batch, different combinations of samples were used for biomaterial extraction. All batch information was noted and used as metadata in exploration of the sources of variability in the data analyses. We noted that two samples (GM12877 and GM12893) grew at a slower rate than the other 15. We also checked carefully whether GM12889 was an outlier in any of the functional assays, as we have found this cell line to have a previously unrecognized mosaic copy number neutral loss of heterozygosity of distal chromosome 11[43], but did not observe any distinctive results from this sample.

We performed a set of genome-wide functional assays on the 17 samples. Directional RNA-seq was used for gene expression profiling, small RNA-seq to test miRNA expression, and the assay for transposase-accessible chromatin (ATAC-seq) to map nucleosome-free, *cis*-regulatory regions[42]. We also performed exploratory whole-genome bisulphite sequencing (WGBS) assays using our optimized Illumina HiSeq X-based assay[44] to profile DNA methylation throughout the genomes of these cell lines.

Our analytical studies started by testing for sources of variability of our results. One of the cell lines that were growing more slowly than the others, GM12893, in addition to batch-affected lines GM12880 and 12879 were also outliers in terms of their gene expression (Supplementary Fig. 2). We used our alignments to the EBV genome to test for differences in expression from the virus, identifying a specific pattern within GM12893 (Supplementary Fig. 3) that includes the *BHLF1* gene, which is expressed early in the lytic cycle of EBV infection and expressed in subclonal populations of LCLs[45]. DNA sequence variants found in these assays were compared with those from the Illumina Platinum Genomes annotations[41] to confirm sample identities. The influences on our functional assays are summarized in principal component plots in Supplementary Fig. 4.

We then focused analyses on the 11 children in third generation of the family, with the goal of identifying functional genetic variants. We used linear regression to account for the identified sources of variability and to generate data more likely to reflect the genomic properties of each cell sample. The published haplotype block data on these Platinum genomes[41] allowed us to test whether gene or miRNA expression in each of the 11 children was associated with paternal or maternal inheritance of the surrounding haplotype, an approach used previously for identifying expression quantitative trait loci (eQTLs) in this family[46]. This allowed us to compare the target genes affected by eQTLs (eGenes) between our study and the prior analysis of this family. We found 21 eGenes in common between the studies (Supplementary Data 1, Fig. 1), a statistically-significant degree of

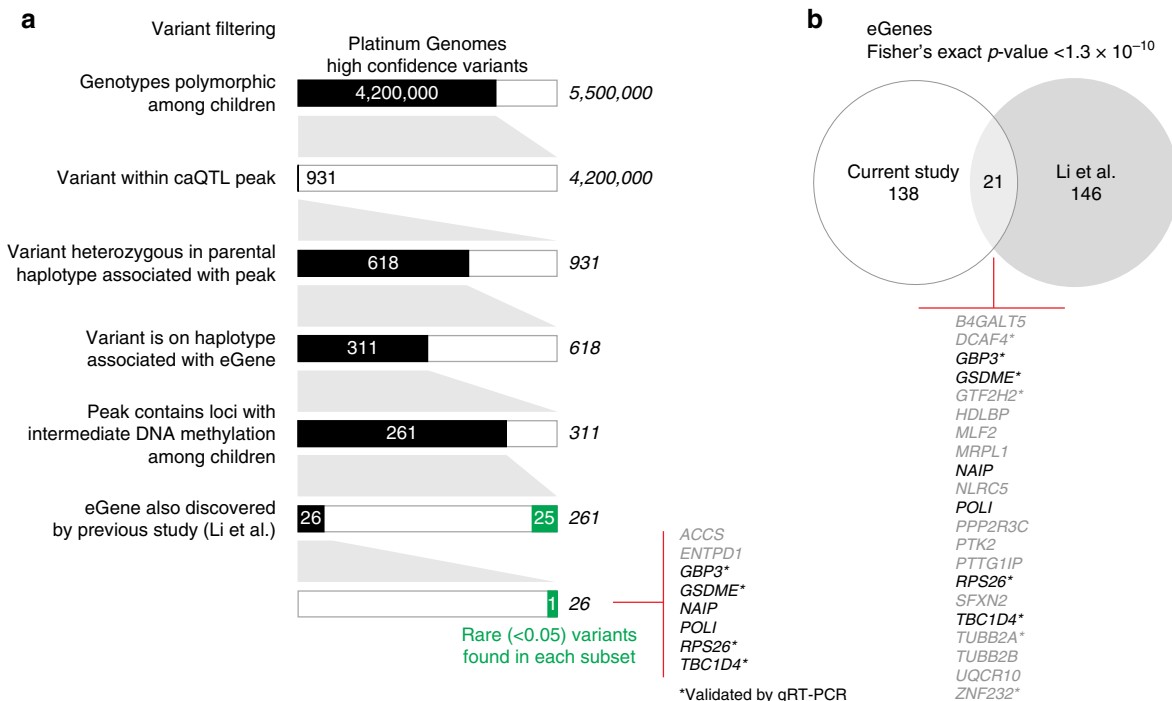

**Fig. 1** Identification of high-confidence multifunctional variants. In **a** we show the steps involved in proceeding from 5.5 million high-confidence variants in this family to the 26 likely to have effects on gene expression, chromatin accessibility, and DNA methylation. An ATAC-seq peak for which the number of reads is influenced by a specific parental haplotype was required to include a polymorphic sequence within the peak, a polymorphism for which that parent had to be heterozygous. We then further filtered by requiring the locus to have an intermediate DNA methylation value, consistent with only one allele being active, and then filtered to overlap with the eGenes that we found to be in common with Li et al.[46] This list is shown in **b** with those validated by qRT-PCR marked with an asterisk, and those genes in common to (**a**) and (**b**) shown in black. We proceeded with *TBC1D4* for further testing

overlap ($p < 1.3 \times 10^{-10}$) despite the prior study's older haplotype annotation and their use of an mRNA-seq assay[46] as opposed to our RNA-seq assay that was not solely dependent upon oligo-dT priming. We further validated the results from both studies by performing qRT-PCR on several eGenes and found that those discovered by both studies almost universally showed the predicted expression differences among the children (8 out of 9) (Supplementary Data 2).

We were concerned that a functional variant might be acting on the expression of a microRNA, which could then be acting on a gene to alter its mRNA levels, leading to the impression that the functional variant was acting on the gene directly. We therefore performed microRNA sequencing to a substantial depth (average of 48 million reads per sample) to allow sensitive identification of all miRNAs being expressed, pooling reads across all 17 samples for prediction purposes. We show there to be 632 known human miRNAs expressed in these samples, plus 13 from the EBV genome, and we identified 112 candidate novel human miRNAs. We confirmed one of these novel human miRNAs using stem-loop PCR (Supplementary Fig. 5), and performed down-sampling to reveal that the number of reads needed per sample to identify this novel miRNA through prediction software was high, estimated at 500,000,000, showing how miRNA discovery may require unexpectedly deep sequencing approaches. We used the same haplotype-based approach to identify eQTLs for the human miRNAs, finding 47 miRNAs whose expression appeared to be influenced by their parental haplotypic inheritance (Supplementary Data 3). We then tested to see whether any of the *cis*-eQTLs we had identified for genes were, in fact, mediated by miRNAs acting upon and located within the sample haplotype block as the eGene, and found no evidence for this in our data set.

The ATAC-seq data were then tested for functional loci influencing chromatin accessibility (caQTLs). Analytically, we mapped ATAC-seq reads with the steps of extra stringency to exclude blacklisted regions of the human genome[47] and loci with sequences similar to mitochondrial DNA, as these sequences are highly represented in ATAC-seq data and are potentially prone to mis-mapping[48]. We combined the reads from all 17 individuals to create a data super-set, with which we mapped ATAC-seq peaks, aiming to define all possible high-confidence peaks in the LCL genome and the consensus positions of the summits within each peak (Supplementary Data 4). We then called peaks individually in each of the 11 children, and identified peaks also present in the super-set that were found in at least 3 of the children. The summit regions within these filtered peaks were found to be enriched for B lymphocyte-specific TFs (Supplementary Data 5). We tested whether the normalized number of reads at each summit region (the summit nucleotide ± 250 bp) exhibited parental haplotype dependence, defining them as the targets of sequence variation influencing chromatin accessibility, or caSummits. We found 276 caSummits in total, loci significantly enriched for containing a heterozygous SNP.

Having identified these individual eGenes and caSummits, we then narrowed the focus to define candidate multifunctional variants affecting both gene expression and chromatin accessibility. We identified 62 haplotype blocks containing both an eGene and a caSummit which overlaps a heterozygous SNP. We also filtered to require that there be co-variation of the informative haplotype and SNP. For example, if the peak is associated with one but not the other paternal haplotype, but the SNP variation is carried on the maternal haplotype, the variation in chromatin accessibility cannot be attributed to that

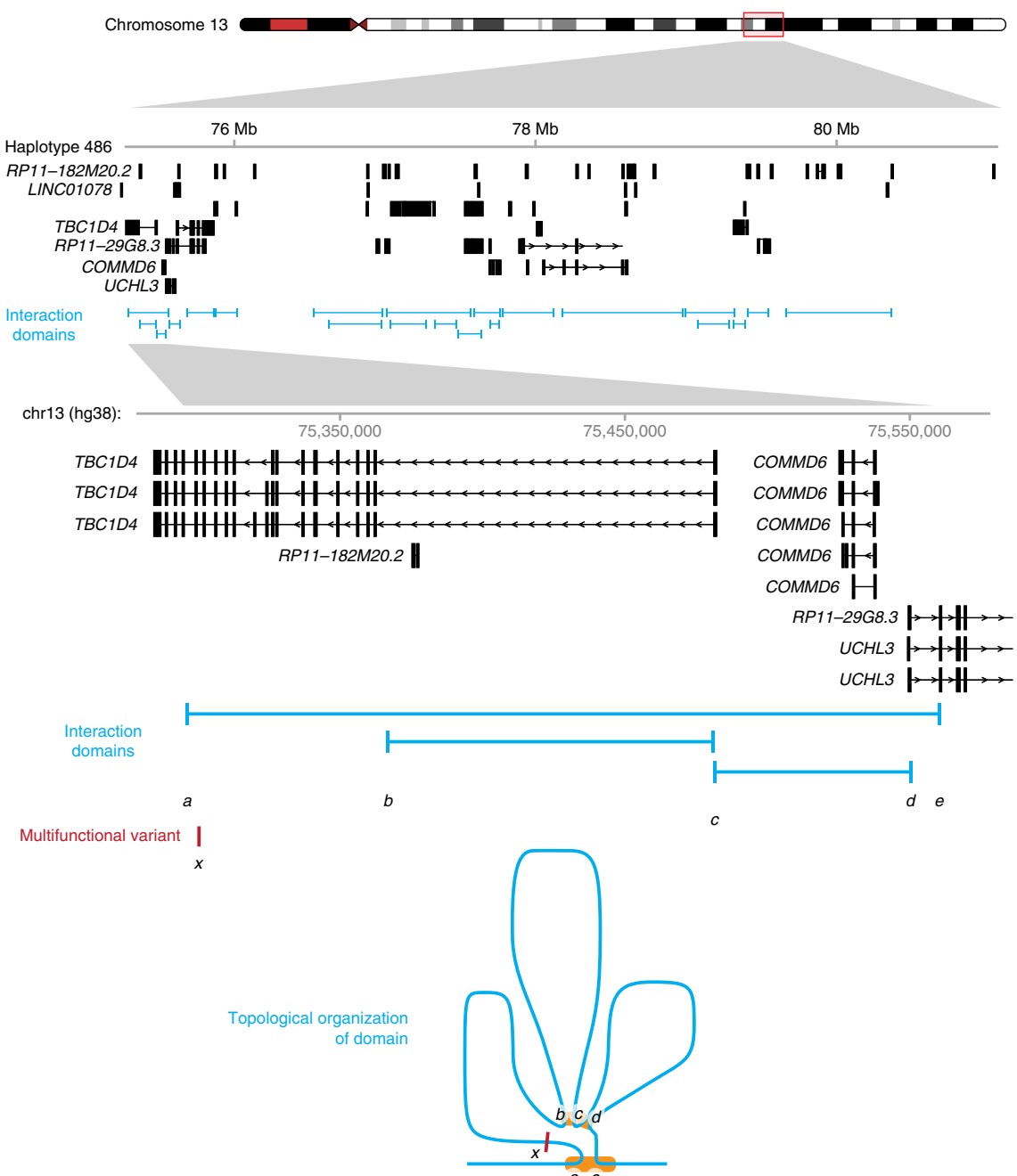

**Fig. 2** Overview of the *TBC1D4* haplotype. The *TBC1D4* gene is at the centromeric end of an ~6 Mb haplotype on chromosome 13q. The genes and three-dimensional interaction domains are depicted, with the lower part of the panel showing the detail in the immediate *TBC1D4* region. Three promoters are located within the major loop containing *TBC1D4*. We show how the promoters and the multifunctional variant are predicted to be organized in three dimensions in vivo, revealing a relative proximity of the multifunctional variant (x) to the *TBC1D4* (c) and *UCHL3* (d) promoters, despite the multifunctional variant being 182 kb centromeric to the *TBC1D4* promoter. Source data of the interaction domains are provided in a Source Data file

particular SNP within the paternal haplotype. As a further filter, with the assumption that one allele being active and the other silenced should be associated with a pattern of intermediate DNA methylation locally, we chose the subset of loci in which CG dinucleotides with DNA methylation values of 20–80% were found within the peak. The overall filtering process is summarized in Fig. 1. We were left with a total of 261 candidate multifunctional variants, including 25 with a minor allele frequency (MAF) of <0.05 in any superpopulation (including Caucasians), demonstrating how some rare variants can be found to be functional when families are studied.

**Characterization of the TBC1D4 multifunctional variant**. The list of high-confidence candidate multifunctional variants included sequences within the *GSDME* and *TBC1D4* genes. We focused on the *TBC1D4* multifunctional variant, located in a ~6 Mb haplotype on chromosome 13q (Fig. 2, upper), an intronic site ~182 kb from the gene promoter that is predicted to loop to relative proximity to the promoter *in vivo*[49] (Fig. 2, lower). The variant consists of a haplotype of 3 sequence changes within 7 bp that are found only in *Homo sapiens* and not other hominids or primates (Fig. 3a). The Geography of Genetic Variants (GGV) browser shows the human-specific allele to be the less common in

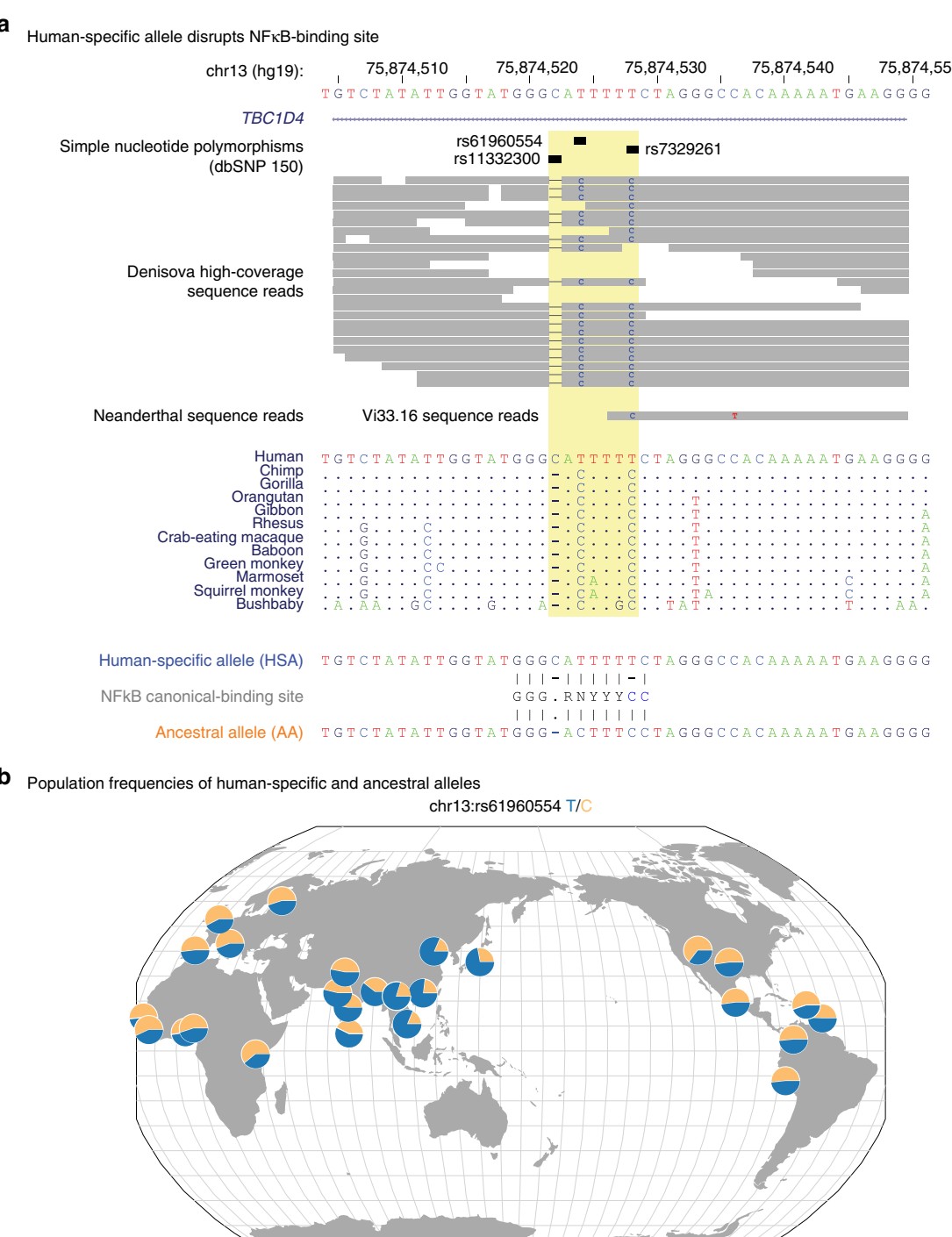

**Fig. 3** Characterization of the multifunctional variant at the *TBC1D4* locus. **a** Comparative sequence analysis shows three variants within 7 bp, the insertion of a C and two C>T transitions that are only found in *Homo sapiens*, not in any other primate or Denisovan DNA, while the one Neanderthal read at the locus also shows the ancestral C. The ancestral haplotype encodes a canonical NFκB binding site, with the human-specific haplotype disrupting it in three locations. In **b**, using data from the GGV browser[50], we observe that while the ancestral allele is the more common in European and African populations, in South and East Asia the human-specific allele becomes substantially more common in these populations

Northern Europeans and Sub-Saharan Africans, but to be the dominant allele in East Asian populations (Fig. 3b)[50]. Using HaploReg[51], we find the ancestral allele (AA) to encode a canonical NFκB binding site[52], with the human-specific (HS) allele changing that binding site in 3 positions, alterations strongly predicted to disrupt NFκB binding. We therefore chose this as the locus for genomic editing experiments. *TBC1D4* is a gene

encoding a Rab-GTPase-activating protein that responds to insulin by dissociating itself from the glucose transporter 4 (GLUT4) protein, allowing the latter to migrate to the plasma membrane where it can act to transport glucose into the cell[53]. Mutations of *TBC1D4* have been associated with type 2 diabetes mellitus[54,55], indicating that the protein has a role to maintain normal glucose homeostasis.

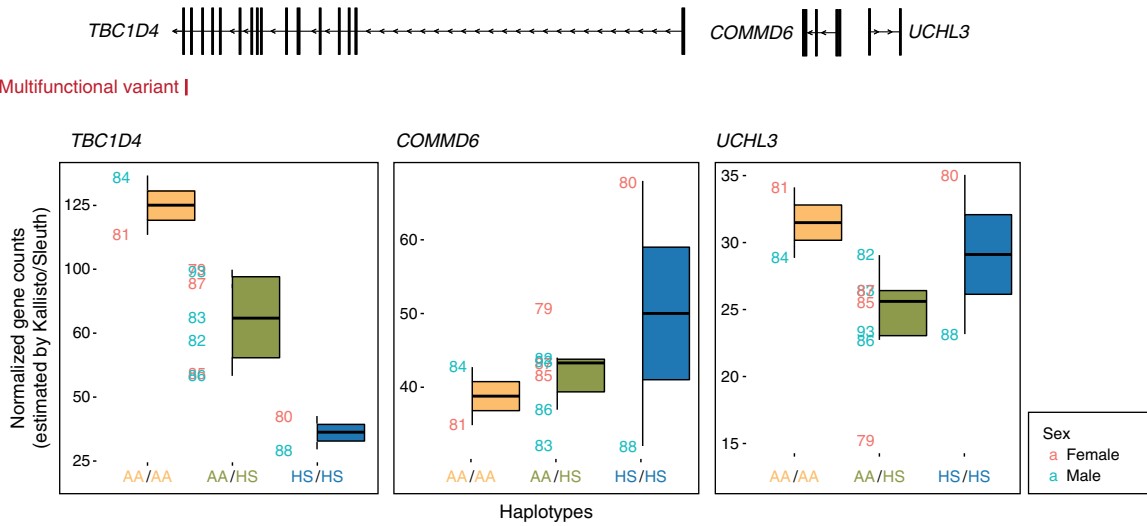

**Fig. 4** Expression of genes within local contact domain. The intronic *TBC1D4* cis-regulatory locus encodes an expression quantitative trait locus (eQTL) that acts on the *TBC1D4* gene but not the immediate upstream *COMMD6* or *UCHL3* genes. For *TBC1D4*, the amount of expression is higher when the ancestral allele (AA) is present compared with the human-specific (HS) allele. Boxplots represent the interquartile range (IQR) with the median denoted by the middle line; whiskers extend to the largest/smallest value no further than 1.5 times the IQR. Expression values are provided in a Source Data file

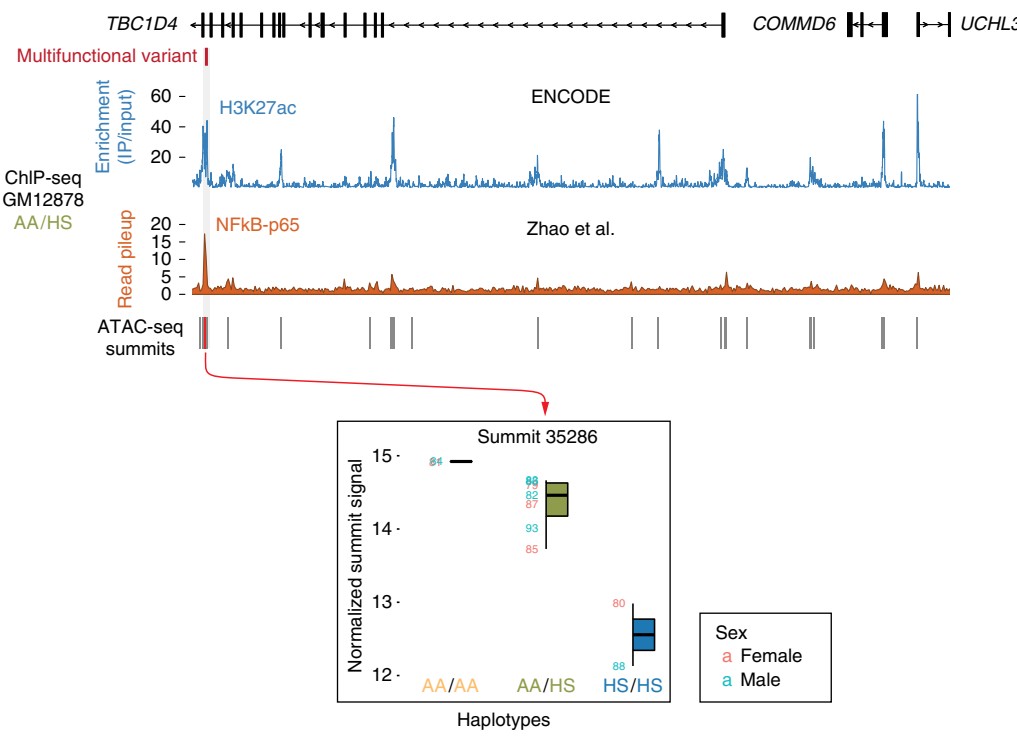

**Fig. 5** Chromatin accessibility within the *TBC1D4* multifunctional variant interaction domain. The intronic *TBC1D4* cis-regulatory locus encodes a chromatin accessibility quantitative trait locus (caQTL) that is located at a site of strong ChIP-seq enrichment for H3K27ac and of read pileup for the p65 component of NFκB, but is only significantly associated with the overlying ATAC-seq peak, with a higher normalized read count (summit signal) associated with the ancestral allele (AA) compared with the human-specific (HS) allele. Boxplots represent the interquartile range (IQR) with the median denoted by the middle line; whiskers extend to the largest/smallest value no further than 1.5 times the IQR. Summit signal values are provided in a Source Data file

The RNA-seq data we generated show that the individuals with the AA (capable of binding NFκB) have higher levels of expression of *TBC1D4* but not the adjacent *COMMD6* or *UCHL3* genes within the same interaction domain (Fig. 4). Our ATAC-seq data for this region show only the summit immediately overlying the variant to be associated with an effect of haplotype inheritance (Fig. 5). More detailed scrutiny of the local chromatin accessibility peak (Fig. 6) shows that the reads generated by ATAC-seq in a heterozygous individual are all from the AA and none from the HS allele (Fig. 6b). In the heterozygous GM12878 cell line, chromatin modifications indicating the presence of an active enhancer (H3K27ac) are enriched at this locus, with binding of NFκB, and the majority of ChIP-seq reads for NFκB components in the heterozygous AA/HS GM12878 cell line[56]

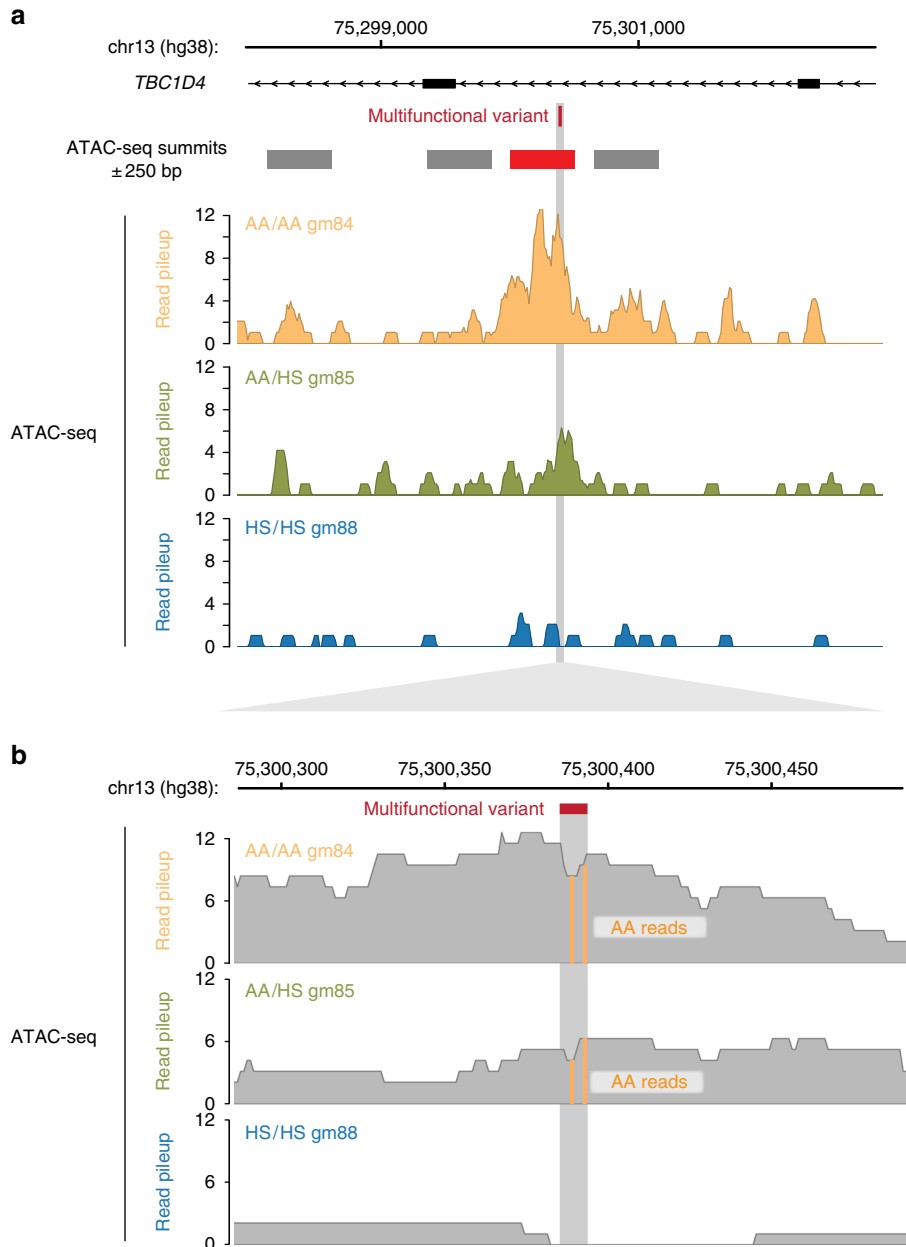

**Fig. 6** Allelic chromatin accessibility at the *TBC1D4* multifunctional variant. The *TBC1D4* multifunctional variant is associated with open chromatin present on the ancestral allele (AA) but not on the human-specific (HS) allele. In **a** we show the 5 kb region flanking the local ATAC-seq peaks, with the peak (summit ± 250 bp) reflecting the caQTL shown in red. In **b** we represent polymorphic AA sequences where C>T transitions are present with orange, showing that in the heterozygous GM(128)85 cell line, all of the reads are from the AA, and none from the HS allele in the same cell

coming from the AA allele, in which the NFκB motif is maintained (Fig. 7).

As there are no informative CG dinucleotides immediately adjacent to the multifunctional variant, we are unable to test haplotype-specific DNA methylation patterns. However, we observe that DNA methylation in immediately flanking CGs is lower in individuals with the AA haplotype (Fig. 8, Supplementary Data 6).

**Genomic editing of the TBC1D4 multifunctional variant**. We proceeded to design an experiment to edit the haplotype at this locus to test the functional consequences of altering the DNA sequence at a predicted TFBS. We chose to perform the editing in LCLs because the GTEx data describing this *cis*-regulatory

locus show it to be specifically active in LCLs[57] (Supplementary Fig. 6). We describe how we overcame the challenges of performing CRISPR-Cas9-mediated genomic editing in LCLs in detail in a companion publication[58], employing a transient transfection protocol using a single-stranded donor oligonucleotide (ssODN) strategy for 'scarless' editing[59,60]. It has recently been highlighted that CRISPR-mediated genomic editing can be associated with mutations that can be large in size and potentially missed if the screen is solely focused on a small, targeted desired mutation[61]. We, therefore, added long-range PCR of the 4 kb DNA region flanking the targeted locus as well as reverse transcriptase PCR (RT-PCR) of overlapping 1 kb segments of the 6.4 kb *TBC1D4* transcript which spans 197.4 kb on chromosome 13. Of the 5 clones found to have edits

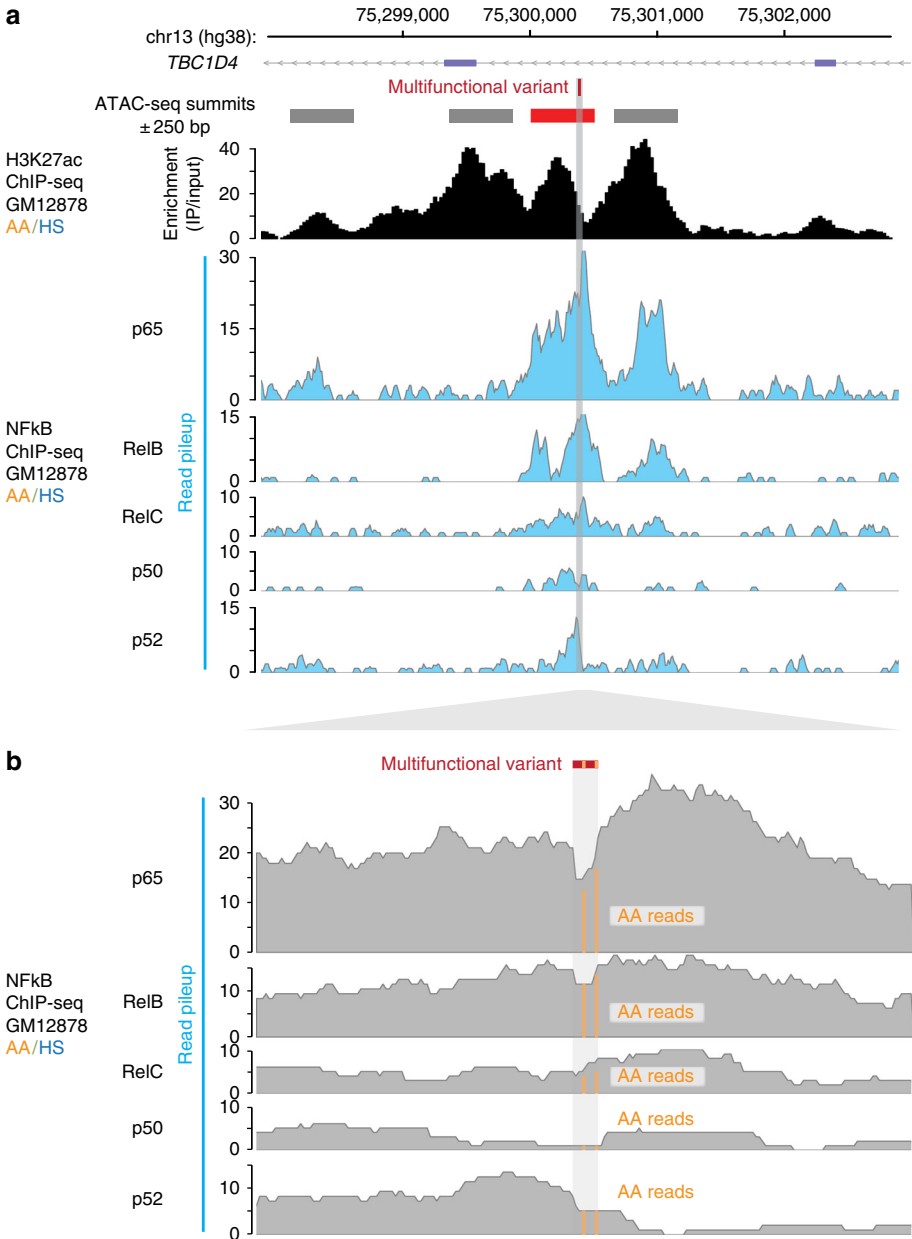

**Fig. 7** Allele-specific binding of NFκB at variant locus ChIP-seq read pileups for NFκB components p65, RelB, RelC, p50, and p52 from the heterozygous AA/HS GM12878 cell line, showing the 5 kb context above (**a**), and a ~200 bp detail (below; **b**) showing most reads of each NFκB component are from the AA

at the desired site, these PCR-based analyses revealed 2 clones to have large mutational events at the locus by testing DNA and RNA[58]. We, therefore, focused our analyses on 3 clones, 2 of which had the desired clean replacement of the Ancestral (AA) with the human-specific (HS) allele, and one with a 7 bp deletion spanning the NFκB binding site that appeared by PCR and densitometry to be hemizygous[58], an even more deleterious pair of mutations for comparison (Fig. 9a).

We performed quantitative RT-PCR (qRT-PCR) to test the effect of replacing the functional, NFκB-binding AA with the inactive HS allele or the cell line with the completely deleted NFκB binding site. We show the results in Fig. 9. A consistent decrease in expression of *TBC1D4* is observed in all edited cell lines (Fig. 9b). We then tested whether this genomic editing did indeed lead to decreased chromatin accessibility and abolition of

NFκB binding by performing quantitative ATAC-seq and quantitative ChIP at this locus. The results show strong decreases in chromatin accessibility (Fig. 9c) and the binding of NFκB (Fig. 9d) at the edited locus. We conclude that these experiments demonstrate a strong association between TF binding at a *cis*-regulatory locus and gene expression, as has been found in numerous prior studies[12–15].

**Testing the role of TFs using CRISPR-mediated re-targeting.** To perform a direct test of the hypothesis that TF binding mediates the transcriptional regulatory effects at functional variants, we again targeted the CRISPR system to the locus, but this time replacing the Cas9 with a nuclease-deficient version, to which we fused the p65 component of NFκB. We used a guide RNA that targeted a site immediately flanking the edited NFκB

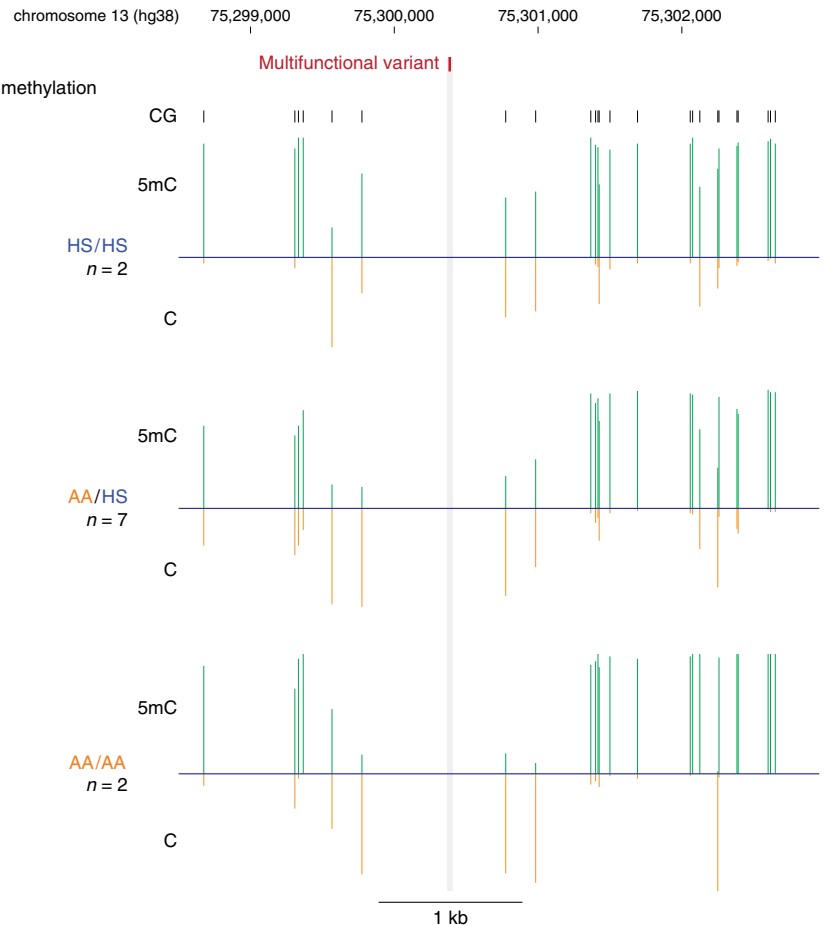

**Fig. 8** DNA methylation in the 5 kb context of the *TBC1D4* multifunctional variant. We show the DNA methylation (5mC) for each cytosine in green with the corresponding unmethylated (C) fraction of alleles in orange, so that loci found to have very low levels of DNA methylation do not appear to be missing in the representation. We show the mean values for the 2 HS/HS, 7 AA/HS, and 2 AA/AA individuals. There are no cytosines sufficiently close to the multifunctional variants to allow observation of haplotypic patterns of DNA methylation, but there is a pattern of less DNA methylation at loci closest to the multifunctional variant in individuals with the AA haplotype. 5mC values, along with median coverage, are provided in a Source Data file

site. We show in Fig. 10a that this strategy was enough to increase *TBC1D4* activity >2-fold relative to a condition in which the same system without the p65 protein was introduced into the LCLs with the HS allele. However, we were concerned that these effects could be partially or wholly due to the increased amount of this NFκB subunit in the cell, binding to other *cis*-regulatory sites at this gene. We, therefore, employed two additional strategies. We developed a control system to target a known NFκB binding site at the *OXTR* gene[62], validating targeting by testing the Cas9-induced mutation rate at this locus in LCLs (Supplementary Figure 7). Our goal was to use this chromosome 3 locus as a 'sink' for the p65 expressed in the cell, leaving less available for ectopic activity at *TBC1D4*. We also removed the DNA-binding domain of p65, preserving the transactivation domain needed to recruit transcriptional regulatory factors[63,64]. Using this strategy, we tested *TBC1D4* expression in a clone homozygous for the HS allele and another clone with the 7 bp deletion disrupting NFκB binding, comparing the expression levels of constructs targeting the edited *TBC1D4* locus and those targeting the *OXTR* locus on chromosome 3. In Fig. 10b we show that both the HS allele and 7 bp deletion cell lines had their levels of *TBC1D4* expression increased by targeting only the functional domain of p65 to a locus that could no longer bind this component of the NFκB TF.

## Discussion

The results of this study lend support to the commonly-held hypothesis that the mechanism by which functional variants exert their effects is through the sequence variability leading to altered binding of TFs. This does not exclude a role for genetic polymorphism at these loci modifying enhancer RNA structure and altered chromatin looping[16], and there is certainly room for more detailed studies of genetic effects on transcriptional regulation, but this study is, to our knowledge, the first that dissociates the role of TF binding from the presence of its cognate motif, moving beyond the strong associations that have been made to date and providing direct evidence to support the TF mediation mechanism.

The interest in identifying functional variants has arisen from associations implicating haplotypes in disease heritability[65]. Genome-wide association study (GWAS) drove a need to identify 'epigenetic' marks such as DNase hypersensitivity, or ChIP-seq mapping of active chromatin regulators, demonstrating enrichment for overlap of these candidate regulatory loci with the lead SNPs identified in the GWAS, finding even greater enrichment for other SNPs in linkage disequilibrium[65]. To progress from these relatively large areas of the genome, containing many candidate functional variants, to the specific variants mediating the disease, the approach has generally been to identify functional variants within these regions. Typically these variants are defined

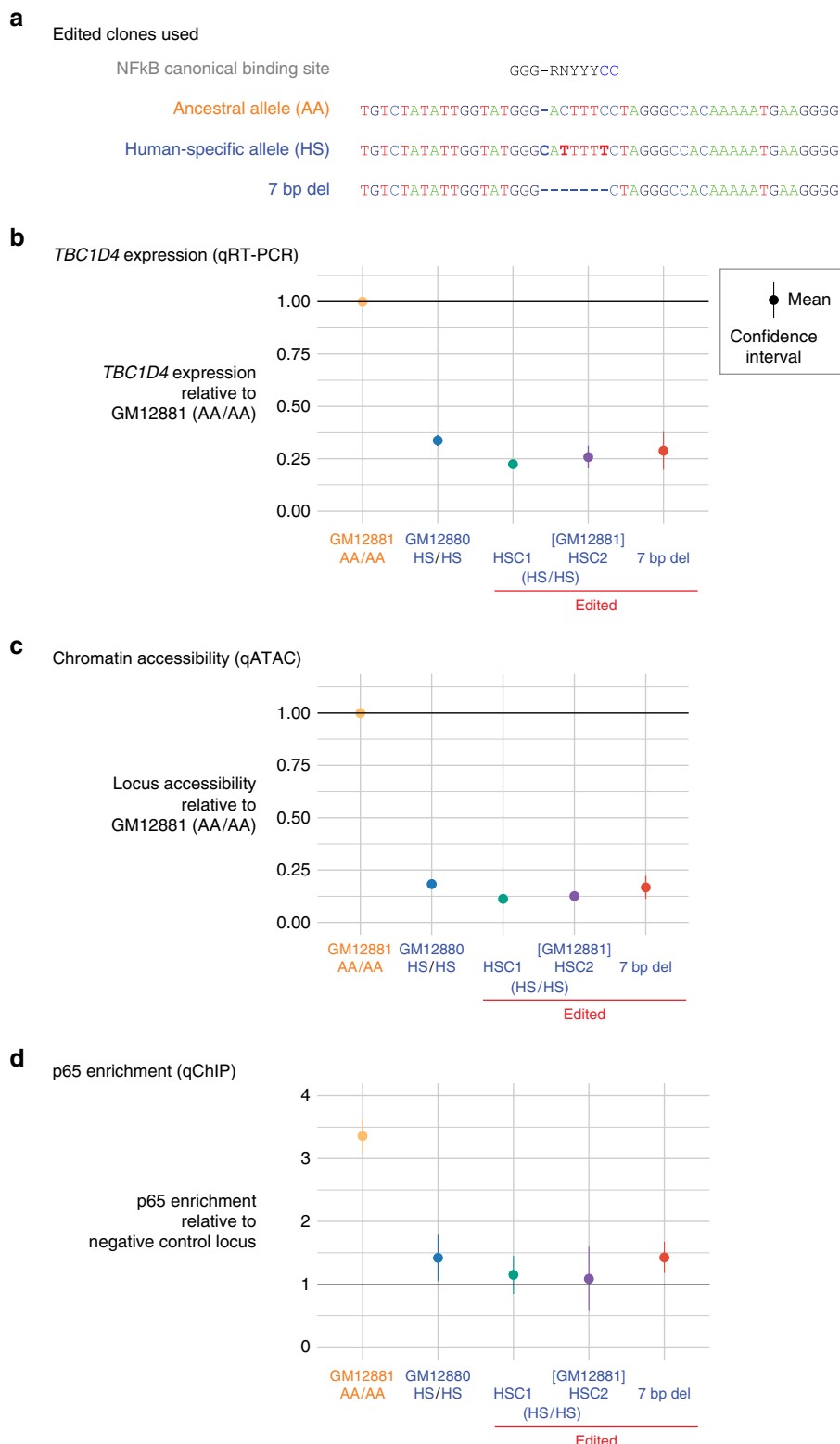

**Fig. 9** Allelic editing leads to local functional changes among edited clones. In **a** we show the ancestral and human-specific alleles, and a 7 bp deletion that was also generated by CRISPR-Cas9 editing. In **b** *TBC1D4* gene expression levels are shown in the unedited GM12881 (AA/AA) and GM12880 (HS/HS) LCLs, compared with two human-specific clones (HSC1, HSC1) and the 7 bp deletion clone generated from GM12881 ($n = 3$ independent experiments each). As expected, there is a strong decrease in expression of *TBC1D4* in the HS/HS GM12880 LCL compared with AA/AA GM12881, while the edited HSC1, HSC2, and 7 bp deletion clones show levels of expression comparable with the HS/HS unedited control. In **c** we show that this editing is associated with a decrease in chromatin accessibility ($n = 3$ independent experiments each), and in **d** with a decrease in the p65 component of NFκB using quantitative ChIP ($n = 3$ independent experiments each). Source data are provided in a Source Data file

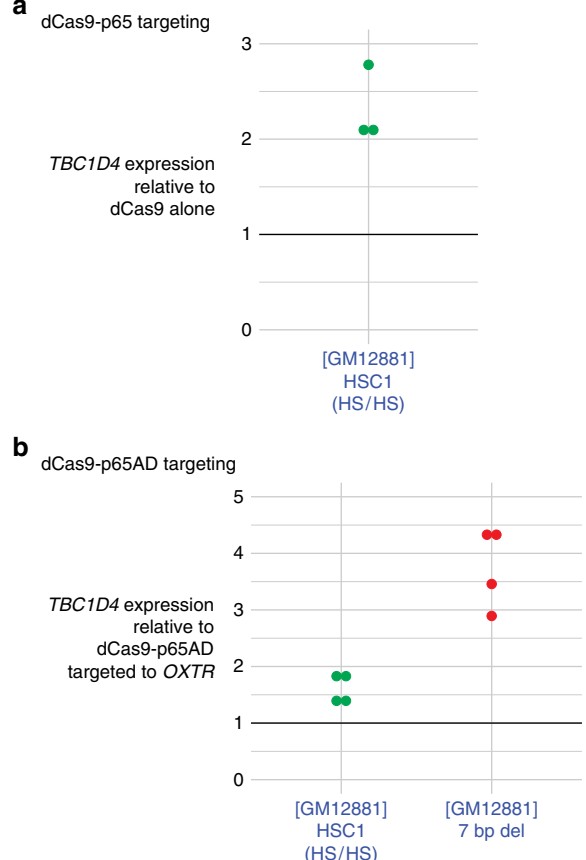

**Fig. 10** Targeting of NFκB to the edited intronic locus helps to restore *TBC1D4* expression. In **a** we show the effect of targeting of the full-length p65 component of NFκB using CRISPR and catalytically-inactive Cas9 to the edited locus (HSC1 cell line), using a guide RNA targeting the edited NFκB binding site. This was associated with the increase in expression levels shown (dots show individual replicates, for clarity). Because we were concerned that this targeting of a locus ~182 kb from the gene's transcription start site (TSS) was working unexpectedly efficiently, and the possibility that we may be having the introduced p65 binding at other NFκB motifs, including the TSS itself, we added two additional measures. We introduced a p65 construct lacking its DNA-binding domain, preserving its trans-activating domains, and used a control guide RNA targeting a known NFκB binding site at the *OXTR* gene, with the goal of having this locus act as a sink for any ectopic p65 in the cell nucleus. In **b** we again show individual results from HSC1 replicates, and also the 7 bp deletion clone as an independent set of replicates. Once again, we see enhancement of *TBC1D4* expression, without the same concerns for ectopic p65 activity. The TF replacement strategy at this locus thus appears to be having effects on transcription acting over a distance of 182 kb, which is unusually far for typical epigenetic editing experiments. Source data are provided in a Source Data file

by associating co-variation of transcription levels with sequence polymorphism *in cis* to that gene, but there are now studies that use DNA methylation[66] or chromatin characteristics[67] to help identify loci with transcriptional regulatory properties. The locus that we studied within the *TBC1D4* gene was selected for its effects on local gene expression, chromatin accessibility and local changes in DNA methylation, as well as its strong predicted effect on the binding of the NFκB TF based on sequence characteristics. This confirms that it should be possible to use different types of functional genomic assays to arrive at the same functional variants, and that the use of multiple functional assays

simultaneously[67] may be especially valuable in prioritizing certain loci as being more likely to have a regulatory function. A functional variant, once identified, can be linked to variable gene expression as an eQTL, with the target eGene representing a potential insight into the mechanism of the disease, based on the properties of the protein encoded by that gene.

One concern has been that the post-GWAS surveys for sequence variants with regulatory functions may have been performed in a cell type irrelevant to the disease process being studied, prompting the search for functional variants in the cell type (s) assumed to mediate the phenotype of interest. That eQTLs can have a cell type-specific activity is increasingly apparent[68], especially apparent from the systematic studies of the GTEx Consortium[69]. The *cis*-regulatory landscape of different cell types is well known to be distinctive[70], implying sequence-specific mechanisms for targeting post-translational modifications of histones, DNA methylation, and nucleosomal remodeling in different cell types. Candidates for mediating this sequence-specificity should include TFs (and possibly regulatory, non-coding RNAs), but not the enzymatic mediators of chromatin states or DNA modifications, which inherently lack the ability to recognize specific sequences. The model of a primary recognition of specific sequences by different TFs as part of the process of driving lineage commitment is well-established. The fact that some TF binding sites (TFBSs) are variable between individuals, therefore, leads to the idea that the same canonical cell type from individuals differing in their DNA sequences at TFBSs can confer distinctive properties on that differentiated cell between individuals, differences that could be reflected by altered or disease phenotypes.

This train of logic implies a primary role for TFs, with secondary effects on chromatin states and DNA modifications. Such a viewpoint raises a challenge to the idea that epigenetic regulation of the genome, defined in terms of chromatin patterns and DNA methylation, acts instead in a primary role to define where TFs can bind within the genome. There is support for both possibilities, mouse embryonic stem cell studies showing an instructive role for TFs on DNA methylation[71] and for DNA methylation influencing the genomic binding of at least one TF, NRF1[72]. There is clear evidence from in vitro studies for DNA methylation influencing TF binding, both negatively and positively[73]. The question therefore arises whether our targeting of the p65 transactivation domains to a specific locus, an example of epigenetic editing, is exerting its positive effects on *TBC1D4* transcription by means of replicating the function of the TF no longer capable of binding to the allele lacking the NFκB motif, or whether it is instead creating a local environment conducive to TF binding in general. As the targeting of the CRISPR complex is mediated through a guide RNA that forms an RNA:DNA hybrid[74], loci which we have shown to be mostly inactive in the human genome[75], and prior studies have revealed NFκB to bind strongly preferentially to double-stranded DNA[76], we believe that, if anything, the targeting of the locus by CRISPR would have an inhibitory effect on local TF binding. While formally unproven by our studies, the primary mechanism for activation of the locus would appear to be more likely to be by the restoration of p65 than by non-specific effects on local chromatin organization. The question, therefore, arises about epigenetic editing more broadly, whether targeting of a locus to induce what may be secondary consequences of local TF binding (histone acetylation, DNA demethylation) is the optimal approach. If the TF (or group of TFs) that would normally bind locally and recruit the mediators of these activation marks is not expressed in that cell, will transcription still be promoted? The different viewpoints inherent to the epigenetic editing and functional genetic communities could end up, even

inadvertently, prompting some valuable insights into transcriptional regulation.

In summary, this study follows up on a wealth of excellent association data indicating a primary role for TFs in mediating effects of sequence variability at functional genetic loci. We harnessed the genetic and epigenetic editing capabilities of the CRISPR-Cas9 system to provide a direct test of this hypothesis, by engineering a locus to fail to bind the cognate TF, then restoring its targeting locally by guiding the transactivation domains of the TF back to the locus using CRISPR, increasing the transcription of the *TBC1D4* gene from a promoter located ~182 kb telomerically.

## Methods

**Cell line culture**. Lymphoblastoid cells (LCLs) derived from the 17-member CEPH Pedigree 1463 were purchased from the Coriell Institute and cultured in RPMI 1640 medium, supplemented with 15% fetal bovine serum (FBS, Benchmark), 100 IU/ml penicillin, and 100 μg/ml streptomycin (Life Technologies). Cells were kept in suspension in tissue culture flasks (NUNC, Thermo Scientific) at 37 °C in a 5% $CO_2$ incubator and maintained between $2 \times 10^5$ and $8 \times 10^5$ cells/ml.

**ATAC-seq library preparation**. The assay for transposase-accessible chromatin (ATAC)-seq libraries were prepared similarly to Buenrostro et al.[42]. For each cell line, we collected $5 \times 10^5$ cells from two different cultures, harvested during the exponential growth phase. The cells were pelleted at 500×*g* at 4 °C for 5 min and washed with 50 μl of cold PBS. Samples were then centrifuged again at 500×*g* at 4 °C for 5 min. Cells were lysed in ice-cold lysis buffer (10 mM Tris-HCl, pH 7.4, 10 mM NaCl, 3 mM $MgCl_2$ and 0.1% NP40) and immediately spun at 500×*g* at 4 °C for 5 min. The pellet was then resuspended in 50 μl of the transposase reaction mixture (TD buffer supplemented with 100 nM transposase). Following a 30-min incubation at 37 °C, the samples were purified using the DNA Clean and Concentrator-5 purification kit (Zymo). After purification, the libraries were amplified and indexed by combining: 10 μl transposed DNA, 5 μl PCR Primer Cocktail (PPC, Illumina FC-121–1030), 15 μl Nextera PCR Master Mix (NPM, Illumina FC-121–1030), 5 μl each of Nextera i5 and i7 indexed amplification primers (Illumina, FC-121–1011), and 10 μl of nuclease-free water. The PCR reaction was carried out using the following conditions: 98 °C, 30 s; a total of 10 cycles of 98 °C, 10 s; 63 °C, 30 s; 72 °C, 1 min; 72 °C, 5 min. Subsequently, the libraries were purified using Agencourt AMPure XP beads; large fragments were filtered by using 0.6× magnetic bead volume (relative to the PCR mixture volume) and then keeping the supernatant. Primer-dimer and short fragments were removed by collecting bead-associated DNA in a 1:1 (bead solution volume:mixture volume) mixture. The libraries were quantified by Qubit HS DNA kit (Life Technologies, Q32851) and their quality assessed by 2% agarose gel electrophoresis and Bioanalyzer High-Sensitivity DNA Assay. Libraries were sequenced on an Illumina HiSeq 2500 to obtain 100 bp paired-end reads.

**XWGBS library preparation**. Whole-genome bisulfite sequencing (WGBS) libraries for the Illumina HiSeq X (XWGBS) were prepared from each cell line similar to the method that we have previously described[44]. A total of $2 \times 10^6$ cells was harvested, pelleted at 2000×*g* for 10 min and frozen at −80 °C until extraction of genomic DNA. Thawed cells were resuspended in 10 mM Tris pH 8.0, 150 mM EDTA solution before the addition of SDS to a final concentration of 1% in a 250 μl volume. Protein was digested overnight at 50 °C with 25 μg of Proteinase K (20 mg/mL, Invitrogen, 25530049), followed by an hour incubation at 37 °C with 5 μg of RNAse A (Sigma). Extraction of genomic DNA was then achieved through two consecutive phenol:chloroform steps, selecting the aqueous phase. The DNA was concentrated and purified via ethanol precipitation. Libraries were then prepared using 100 ng of genomic DNA and an additional spike-in of 0.5 ng of unmethylated lambda DNA as previously described[44]. Libraries were 250 bp paired-end sequenced on the Illumina HiSeq X system.

**Total RNA extraction and dirRNA-seq library preparation**. Cell pellets were treated with QIAzol lysis reagent (Qiagen) and total RNA was isolated using the miRNAeasy kit (Qiagen) combined with on-column DNase (Qiagen) treatment according to the manufacturer's instructions. All dirRNA-seq libraries were prepared by removing rRNA from 1 μg of total RNA and generating strand-sensitive reads with a KAPA Stranded RNA-Seq Kit with RiboErase (HMR), following manufacturer's protocol (Kapa Biosystems). Total RNA was depleted of rRNA by hybridization of complementary DNA oligonucleotides, followed by treatment with RNase H and DNase. RNA was then fragmented using heat and magnesium and submitted to first strand cDNA synthesis using random priming. After second strand synthesis and incorporation of dUTP, the ds-cDNA was A-tailed, ligated with adapters, and the library was amplified with primers carrying appropriate adapter sequences. The libraries were 100 bp paired-end sequenced.

**Small RNA-seq (sRNA-seq) library preparation**. Libraries were constructed using NEBNext Small RNA Library Prep Kit (NEB). Input total RNA was ligated with 3′ adapters, and the excess of 3′ adapter was hybridized with a reverse transcription primer prior to the ligation of the 5′ adapter. Ligated RNA was converted into cDNA with a first strand reverse transcriptase, followed by PCR amplification. Library size was then selected by loading the PCR product into a 3% agarose dye-free gel of a Pippin Prep system (Sage Science). The libraries were 100 bp single-end sequenced. The metrics for all of the library sequencing performed in this project are shown in Supplementary Data 7.

**Quantitative reverse-transcription polymerase chain reaction (qRT-PCR)**. Synthesis of cDNA was performed with total RNA and SuperScript III First-Strand Synthesis System for RT-PCR (Life technologies) using both oligo$(dT)_{20}$ and random hexamers as primers (except for *TBC1D4* clone validation when only oligo $(dT)_{20}$ were used). All qPCR primers were designed using the NCBI Primer-BLAST web interface[77] (Supplementary Data 8). The quantitative PCR was performed with the LightCycler 480 Sybr Green Master mix (Roche), according to the manufacturer's instructions.

**Novel miRNA validation**. Stem-loop primers (Supplementary Data 8) were pre-incubated at decreasing temperatures (95, 80, 70, 60, 50, 40, 30, and 20 °C), each for 30 s. A 20 μl reaction mixture was then prepared with 2.5 μg of total RNA, 10 pmol of each primer, 1 μl of revertAid M-MuLV reverse transcriptase (Millipore), 0.5 μl of Riboblock RNase inhibitor (Thermo Scientific), and 2 μl of 10 mM dNTPs (Roche) in revertAid reaction buffer (Millipore). First strand cDNA synthesis was conducted with the following thermocycler program: 42 °C, 60 min; 70 °C, 15 min; 4 °C, 5 min. Following this, 1 μl of RNase H was added to the reaction followed by incubation at 37 °C for 20 min. The reverse transcriptase (RT) product was used as a template to amplify both the sRNA U6 (spliceosome, control) and the novel miRNA, using the Fast Start High Fidelity PCR System (Roche) with appropriate primers (Supplementary Data 8). Products were analyzed on an agarose gel and the novel miRNA sequence was confirmed by Sanger sequencing following T/A cloning and amplification steps.

**Custom reference genome for alignment**. A custom reference genome composed of GENCODE v24 primary assembly (hg38), decoy version 1 for GRCh38 (hs38d1), and the Epstein Barr Virus genome, was used throughout the analysis[78]. The latter two components were downloaded from the NCBI assembly portal.

**ATAC-seq summit calling and quantification**. Sequenced reads were aligned to the hg38 reference genome using the Burrows-Wheeler Aligner version 0.7.13 (bwa mem –M –t <N> hg38.idxbase <R2 FASTQ > <R1 FASTQ > )[79]. Uniquely mapped reads were retained using *samtools* v0.1.19, followed by removal of mitochondrial reads by *picard-tools* v1.92[80]. Finally, duplicate reads were removed using both *picard-tools* (MarkDuplicates.jar) and *samtools* (samtools rmdup). Read1 reads were shifted using *bedtools2 v2.26.0*, as previously performed[42]. Leveraging the cell type similarity among the samples, all of the shifted reads were concatenated for each of the two replicate batches. Peaks were called for each replicate using *MACS2* v2.1.0 (macs2 callpeak–nomodel–nolambda -g 3e9–keep-dup 'all'–slocal 10000 -t < INPUT FILE > -n < OUTPUT PREFIX>)[81]. Irreproducible discovery rates (IDRs) were found for the overlapping peaks using the method previously described[82], and peaks with an IDR of less than 0.05 were retained and subsequently filtered by consensus blacklisted and mitochondria homologous regions, creating a 'master' peak list (Supplementary Data 4). In parallel, MACS2 also called summit regions for each concatenated replicate, using the '-call-summit' function. Summit regions were defined as ±250 bp around the summit and were first merged within each replicate. When overlaps occurred, the summit with higher pileup height was retained. Subsequently, summit regions were filtered for those within the master peaks. Next, the filtered summit replicates were merged together. Finally, a master summit list of 500 bp regions was generated after removing blacklisted and mitochondria homologous regions, as performed previously (Supplementary Data 4)[48]. As performed for the concatenated replicate, filtered peaks with an IDR < 0.05 were also discovered for each individual. Master summits, which overlapped peaks present in at least three children, were retained for quantitative trait and motif enrichment analysis. Motif enrichment within the summits used in QTL analysis was performed using the MEME-suite web interface[83]. To quantify the accessibility of these selected summits for each of the children, the number of Read1 reads overlapping the summits were counted and normalized by the total number of reads within summits as well as by conditional quantile normalization to remove GC bias[84]. The values of ATAC-seq signal were then regressed as the summation of inheritance effects of parental haplotypes, similarly to Li et al.[46]: $T_i \sim \mu + \beta_j p_i + \beta_k m_i$, where $T_i$ is the quantitative trait of individual $i$, $\mu$ is the intercept, $m$ and $p$ represent the maternal and paternal haplotype inherited by child $i$, and the effects of the haplotype alleles are reflected by $\beta_j$ and $\beta_k$. Only the haplotypes, in which the summit resides, were tested. caSummits were defined by an FDR of less than 0.5.

**Gene quantification and QTL analysis**. Sequenced reads were pseudo-aligned to GENCODE v24 long noncoding RNAs and protein-coding genes, as well as EBV

genes, using *Kallisto* v0.42.5 (kallisto quant -i –hg38.idx -bias -o $SampleName -b 100 -t 10 $Read1 $Read2)[85]. Estimated counts for genes were determined by *Sleuth*[86]. Genes on the sex and EBV chromosomes were removed and filtered for those with an average count of at least 10 among the children. Genes whose expression levels were associated with parental haplotype allele inheritance were discovered as aforementioned for caSummits except with additional regressors for batch extraction date and quartiled mean EBV expression. eGenes were defined by an FDR of less than 0.4. The statistical significance of the overlap between our eGene set and the previous study's set was assessed using *GeneOverlap* v. 1.16.0.

**Novel miRNA discovery, miRNA quantification, and QTL analysis.** Small RNA sequence reads were aligned to the custom hg38 reference genome and a miRBase v21 human miRNAs gene transcript file[87] using the *STAR* aligner v. 2.5.2b with several parameter modifications specific to miRNAs (STAR --genomeDir hg38 --readFilesIn $Read1 --outSAMtype BAM SortedByCoordinate --out-FilterMismatchNoverLmax 0.05 --outFilterMatchNmin 16 --out-FilterScoreMinOverLread 0 --outFilterMatchNminOverLread 0 --alignIntronMax 1 --outFileNamePrefix $Sample). The aligner soft clips the 5′ ends of reads in order to find proper alignments. As this could lead to erroneous alignments, soft clipped reads were removed using *samtools* as well as the secondary alignments of reads. The primary alignments of all family members were concatenated and passed through miRDeep* for novel miRNA detection[88]. Only 221 putatively novel miRNAs with more than 50 supporting reads and a miRDeep* score greater than 10 were retained. A final set of 112 novel miRNAs were obtained by further filtering against the following miRNA databases: Rfam database v.12[89], GENCODE v24[78], fRNAdb[90], NONCODE 2016[91], and FANTOM[92].

Quantification of miRNAs was achieved by using *subRead* v.1.5.0 featureCounts on all reads uniquely mapped by STAR and a miRBase v21 gtf also containing the 221 putative novel miRNAs[93]. Only miRNAs with at least 5 counts in 4 children were retained and were subsequently normalized using the trimmed mean of M values method[94]. These expression levels were regressed by haplotypic inheritance as previously mentioned to find eQTL-miRNAs (e-miRNAs).

To test if e-miRNA expression was mediating the effect of genetic variation on associated eGenes, the expression levels of eGenes were regressed by shared haplotype block miRNA expression levels, as well as by extraction date and quartile of EBV expression. e-miRNAs whose expression levels significantly (FDR < 0.05) predicted the expression of eGenes were not found to target their respective associated genes by testing validated targets stored in the multiMiR database[95].

**DNA methylation analysis.** To define CpGs with intermediate methylation levels, sequence reads were aligned to the custom hg38 reference genome and methylated and unmethylated counts for CpG were calculated as we have previously performed[44]. Only CpGs with at least ten reads defining its methylation status on either strand were retained. Cytosines with coverage in the 99.9th percentile were removed to mitigate potential PCR artifacts. Finally, only the CpGs for which at least three children had intermediate methylation levels (25–75%) were used for discovering functional variants.

**Variant filtering.** As shown in Fig. 1, variants from the high confidence Platinum Genomes study were first filtered for genetic variants that were polymorphic among the children and were located within the peak of a caSummit. The variant within the caSummit had to be heterozygous within the parent whose haplotype allele was significantly associated with the change in accessibility. Next, variants that resided on haplotypes associated with eGenes and within peaks containing CpGs with intermediate methylation levels were selected. Finally, only variants on eQTL haplotypes also discovered by Li et al.[46] were explored for their potential to disrupt TF motif using HaploReg[51]. Rare variants were those with a minor allele frequency less than 0.05 in all 1000 Genomes superpopulations[96].

**Allele specificity of NFκB subunits.** The ChIP-seq dataset from Zhao et al.[56] was downloaded from the NCBI GEO database accession GSE55105. The fastq files were aligned using the Burrows-Wheeler Aligner version 0.7.13 (bwa mem –M –t < N > hg38.idxbase < R1 FASTQ >)[79] to our custom hg38 genome.

**Quantitative ChIP.** For each tested cell line (GM12880, GM18881, and selected, edited GM12880 clones), a total of $20 \times 10^6$ cells was resuspended RPMI 1640 medium with 1% formaldehyde and incubated at room temperature under shaking for 10 min. The cross-linking was quenched with 0.125 M glycine, and the suspension was pelleted by centrifuging at 800×g for 5 min at 4 °C, then washed with 4 ml of ice-cold PBS and pelleted again. The pellets were then resuspended in Farnham lysis buffer and spun down at 800×g for 5 min at 4 °C. Subsequently, the samples were processed in a Bioruptor using the setting of ON/OFF (30 s/30 s) cycles for 10 min. An average shearing size of 200 bp was obtained for each sample. The sonicated mixture was then spun at 10,000×g for 15 min at 4 °C, and the supernatant was collected. Protein G Dynabeads (ThermoFisher) were added to a 2 ml tube containing 1 mL PBS/BSA, the magnetic beads were washed 3 times to a final volume of 200 μl of PBS/BSA, and 20 μg of anti-p65 antibody (Millipore, cat# 17-10060) was added to each tube. The antibody was coupled to the beads for 3 h at 4 °C. Antibody-coupled beads were then washed again 3 times with PBS/BSA and

then added to the 1 ml chromatin preparation and incubated at 4 °C overnight. The beads containing the immuno-bound chromatin were collected using a magnet, followed by washing 5 times in cold LiCl wash buffer and then washing with 1 ml TE buffer. The supernatant was then discarded and the bead pellet was resuspended in 200 μl IP elution buffer and incubated at 65 °C for 1 h, with vortexing every 15 min to elute the immuno-bound chromatin from the beads. The samples were then spun at 20,000×g for 3 min at room temperature. The supernatant containing the immunoprecipitated DNA was incubated at 65 °C overnight to complete the reversal of the formaldehyde cross-linking and then purified using the DNA Clean and Concentrator-5 kit (Zymo Research). The purified DNA was used as a template for quantitative PCR performed with the Light Cycler 480 Syber Green Master mix (Roche), according to the manufacturer's instructions. Enrichment of DNA at the *TBC1D4* cis-regulatory site was compared to a negative control region located within a gene desert of chromosome 13 (71,132,129-71,132, 279, hg19 coordinates) where GM12878 ChIP-seq showed no H3K4me1, H3K27ac or NFκB enrichment. Enrichment at the promoter of IκBα, a p65 binding site validated by EMD Millipore, was used to assess the quality of the immunoprecipitation.

**Quantitative ATAC.** As performed by to Corces et al.[97], a total of $5 \times 10^4$ cells were pelleted by centrifugation at 500×g for 5 min at 4 °C and resuspended in 50 μl of ATAC-Resuspension Buffer (RSB) containing 0.1% NP40, 0.1% Tween-20, and 0.01% digitonin and kept on ice for 3 min. The suspension was washed with RSB devoid of NP40 or digitonin, and nuclei were pelleted by centrifugation at 500×g for 10 min at 4 °C. Nuclei were resuspended in 50 μl of transposition mixture (TD buffer supplemented with 100 nM transposase, 33% PBS, 0.01% digitonin, and 0.2% Tween-20). The reaction mixture was incubated 37 °C for 30 min with 3 g orbital shaking. Subsequently, the sample was purified with the DNA Clean and Concentrator-5 kit (Zymo) and submitted to qPCR as described for quantitative ChIP. A locus within a gene desert of chromosome 13 (71,132,129-71,132, 279, hg19 coordinates) was used as a negative control. The enrichment of accessible chromatin at the *TBC1D4* locus was normalized to the promoter region of *GIN1*, which was found to be highly accessible yet demonstrated invariable signal strength among the children.

**Plasmids and ssODN template.** Our optimization of the CRISPR system for LCLs is described more fully in a companion paper (Johnston et al. BIORXIV/2018/379461). Plasmid pCAG-eCas9-GFP-U6-gRNA was a gift from Jizhong Zou (Addgene plasmid #79145) and was used in combination with an ssODN template (Supplementary Data 8) for editing purposes. Plasmid pmaxGFP (Lonza) was used as GFP-positive control vector. To generate the dCas9 vectors, plasmid pAC154-dual-dCas9VP160-sgExpression (Addgene plasmid # 48240)[98] was double digested, then purified on agarose gel to obtain a linearized fragment containing both the sgRNA expression and the dCas9 sequences. P2A Reverse complement oligonucleotides were annealed and extended, while the EGFP sequence was amplified by PCR from plasmid pBI-MCS-EGFP (Addgene plasmid #16542)[99] using the nucleotides and primers described in Supplementary Data 8. All fragments were Gibson assembled to provide the scrambled sgRNA-dCas9-P2A-EGFP plasmid. To insert the desired gRNA sequences into eCas9 and dCas9 vectors, reverse complement oligonucleotides containing the 20 nt gRNA target sequence (Supplementary Data 8) were annealed, 5'-phosphorylated and ligated into the linearized vector. The sgRNA-dCas9-p65AD-P2A-EGFP plasmids were obtained by Gibson assembly of the p65 fragment, either full length or (trans)activation domain (AD), into the sgRNA-dCas9-P2A-EGFP plasmid. The sequences coding for p65 (full length and AD) were amplified from plasmid pCMV4 p65 (Addgene plasmid #21966)[63] with sequence-specific primers (Supplementary Data 8).

**Transfection and sorting.** LCLs were passaged at $3.5 \times 10^5$ 48 h and 24 h before transfection. A total of $4 \times 10^6$ GM12881 (or subsequent edited clones) cells were transfected with 33.3 μg of CRISPR-Cas9 plasmids in addition to 0.4 nmol of ssODN template for editing. The same number of cells was transfected with 2 μg of GFP plasmid to provide GFP control cells, while negative control cells received transfection reagents only. Transfections were conducted with the Cell Line Nucleofactor Kit V (Lonza) according to the manufacturer's instructions. After transfection, cells were suspended in medium and incubated overnight under cell culture conditions, then replaced with fresh medium. GFP-positive cells were sorted after 48 h following the transfection. Cells were pelleted, washed twice, and suspended in sorting buffer (Hank's balanced salt solution buffer supplemented with 1% FBS, 100 units/mL penicillin, and 100 μg/mL streptomycin). Cell suspensions were submitted to cell analysis and sorting in a FACSAria II cytometer (BD Biosciences). FACS data were analyzed using FACSDiva software (Becton Dickinson) with gating of single cells using FSC/W and SSC/W, and gating of GFP-positive cells. GFP-positive cells were identified by determining the GFP gating cutoff for control GFP-negative cells, further details are included in Johnston et al.[58].

**Clone isolation upon genomic editing.** Single GFP-positive cells were sorted 48 h after transfection into individual wells of a 96-well plate, containing a mixture of fresh and conditioned medium (1:1), in which the FBS concentration was increased

to 20%. The 96-well plate was incubated for two weeks under cell culture conditions, then the content of each well was transferred to a new 96-well plate with additional conditioned medium. Conditioned medium was obtained from GM12881 cells, cultured in 20% FBS RPMI 1640 for 24 h. The medium was removed without disturbing cells at the bottom of the flask, centrifuged at 792×g, and the supernatant was filtered through a 0.3 micron sterile filter prior to use. When subsequent T7EI assays or qRT-PCR analyses were to be performed, cells were sorted directly into QuickExtract DNA extraction solution or RNAlater, respectively.

**T7 endonuclease I assay (T7EI)**. To verify indel efficiencies of gRNAs targeting CRISPR-Cas9 to the *TBC1D4* or the *OXTR* loci, genomic DNA was isolated from transfected and control cell pellets using QuickExtract DNA Extraction Solution (Epicentre) according to the manufacturer's instructions. DNA was then concentrated by ethanol precipitation. The 1 kb region containing the gRNA targeted region was amplified with forward and reverse primers (Supplementary Data 8) using the Q5 Hot Start High-Fidelity 2X Master Mix (NEB) according to the manufacturer's protocol with 100 ng of the purified total cellular DNA in a 50 µl reaction. Amplification products were isolated using the DNA Clean and Concentrator-5 kit (Zymo Research). PCR product (50 ng) was denatured and re-annealed in a final volume of 13 µl in 1X NEBuffer2 (NEB) using a thermocycler with the following protocol: 95 °C, 5 min; 95→85 °C at −2 °C/s; 85→25 °C at −0.1 °C/s; hold at 4 °C. Hybridized PCR products were treated with 10 U of T7E1 enzyme (NEB) at 37 °C for 60 min in a reaction volume of 20 µL. The reaction was stopped with 2 µL of 0.25 M EDTA, and subsequently analyzed on a 1.5% agarose gel.

**Reporting summary**. Further information on research design is available in the Nature Research Reporting Summary linked to this article.

## Data availability
All genome sequencing data are available from the NCBI Gene Expression Omnibus database under accession number GSE117576. All other relevant data supporting the key findings of this study are available within the article and its Supplementary Information files or from the corresponding author upon reasonable request. A reporting summary for this article is available as a Supplementary Information file.

## Code availability
The code files for all analyses are available at https://github.com/GreallyLab/Johnston_et_al.

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

## Acknowledgements

The authors thank the following Einstein core facilities for their expertise: the Epigenomics Shared Facility, the Flow Cytometry Core Facility, the Computational Genomics Core Facility, and the Genomics Core Facility. Michela Ranieri from the laboratory of Antonio Di Cristofano (Einstein) is also thanked for her help. The feedback of colleagues at the New York Genome Center, especially the laboratory of Tuuli Lappalainen, is also gratefully acknowledged. A.D.J. was supported by Einstein's Medical Scientist Training Program NIH NIGMS T32 GM007288. C.A.S.-P. was supported by the European Union's Horizon 2020 research and innovation programme under the Marie Sklodowska-Curie grant agreement No 750190.

## Author contributions

Project design: J.M.G. Project oversight: M.S. and J.M.G. Experimental design: A.D.J., C.A.S.-P., M.S., and J.M.G. Experiment execution: A.D.J. and C.A.S.-P. Data analysis: A.D.J. and T.V.T. Manuscript preparation: A.D.J. and J.M.G. Manuscript editing and finalization: A.D.J., C.A.S.-P., T.V.T., M.S., and J.M.G.

## Additional information

**Competing interests:** The authors declare no competing interests.

