## [Peer Review File · Nature Communications]

Reviewers' comments:

Reviewer #1 (Remarks to the Author):

Johnson et al tried to answer the role of genomics variants on the TF factor binding site for the gene expression, chromatin accessibility and nearby DNA methylation status. There are two main experiments in this project. First, they aim to identify the polymorphic loci with very strong functional effects on gene expression, chromatin accessibility and DNA methylation. They used lymphoblastoid cell lines from one of the well-established pedigrees called Utah1463. In this part, they performed tremendous job to identify the locus in which 3 SNPs were found to control the TBC1D4 gene expression. Next, they modified those variants by using CRISPR genetic/epigenetic tools to support their hypotheses that TFs binding mediates the transcriptional regulation. If they answer some concerns in the second part, I think this paper will help us to understand the role of TFs and epigenetic (chromatin) changes and their crosstalk for the gene expression.

Major Point:

1. In Figure 3 and Figure 4, they demonstrated the loss of NFkB interaction with their target loci by using sequence-based experiment. To show the direct interaction between the TFs and its DNA-motif, they can perform EMSA staining by using anti-P65 or anti-P50 antibody. Super-shift in the EMSA will prove the physical interaction of the protein-DNA on their identified locus.

2. By performing ATAC-seq, they showed very little chromatin accessibility in the HSA/HSA individual with respect to the AA/AA. First, it will be better to demonstrate the peak on a large scale (100 bp on the both sides) in a supplemental figure. Then, they can look at the peak on the variant site for the well-known epigenetic markers. If H3K27ac mark is deposited in their locus, they can test the role of TF and epigenetic mark for the gene expression in their well-suited model. This will help scientific community to understand the current discussion between two groups working on functional variants and performing epigenetic editing. Recently, Hilton et al (Nat biotech, 2015) established dCas9-P300 system to deposit H3K27 mark on promoter and enhancer sites and Kuscu et al (JMB, 2018) used this system to make denovo mark on non-regulatory sites. If authors of this paper accumulate this histone mark on their target site in HSA/HSA cell line where NFkB cannot bind, they have a chance to see the whether TF binding is necessary for the gene induction or not. If this is applicable, p300 will give more direct relationship for the role of epigenetic editing for gene expression than P65.

3. They found their variant site 182kb away from the promoter of TBC1D4, and it seems that they said these variants can only alter the expression of the gene where the variants exist. In figure 4 and 5, they focus only on the TBC1D4 gene expression as well. It is reasonable to focus on the gene identified by QTL analysis; however, they need to show the effect of their genetic/epigenetic alterations on nearby gene to validate the specificity of the TFs for the transcription regulation of their eQTL Gene.

Minor Points:

Line 100-101: Two slow-grow cell lines (GM77 and GM93) were called as outliers for gene expression, however Supp Fig 2 did not show them as an outlier. In this supp figure, GM79 and GM80 were depicted as an outlier in addition to the GM93.

Line 179: Does ATAC-Seq data in Fig3a belong to only GM78 cell line or multiple cell lines? In figure legends, you mentioned that pileup data have been shown for AA/AA(n: 2), AA/HSA(n: 7) and HSA/HAS(n: 2). GM78 has also not been recognized in the panel b and c of the Figure 3 as an AA/HSA sample.

Line 221: I think "HAS" should be "HSA".

Fig2: There is inconsistency for the genomics positions in the panel A and B. If the red arrow shows the variant, then these variants in panel B should be around 75,300,000. Please check

which version of the human genome (hg) was used for each panel.

Fig3a: zoom-out image of the ATAC-seq profile. (for 100-200 bp around the variant region). It will give us a better idea for the position of the variants on or near the peak.

Fig4c: typo; change the "accessibility " into "accessibility".

FigS3: What is the pinkish bar on the top of the panel. What does color density represent?

FigS5: What is the size of the bands in the marker?

Reviewer #2 (Remarks to the Author):

The manuscript by Johnston and colleagues describes a conceptually simple experiment that demonstrates how the alteration of DNA sequence (a multi-functional variant) corresponding to NFkB binding sites alters NFkB binding and the expression of the TBC1D4 gene. The authors employed high-precision genome editing tools to test this in the LCL cell line and have generated a whopping 85 genomics data sets to support this hypothesis. Such studies are much needed as they can greatly aid in clarifying the function of diverse genomic variants. Nevertheless, it is fair to say that none of the reported findings come across as particularly surprising. Also, the manuscript makes a number of priority claims that are not properly backed up by scientific data and are in my opinion unnecessary (see points 1 and 3). I believe that it would be much more useful to place this work in the context of already published literature and highlight the methodological improvements. The figures also need significant improvements as in many cases it is not clear what is being presented and axis labels are not clearly explained. My suggestions can be found below.

The claim in the abstract: "In this study, we provide the first formal proof..." should be removed. If the authors wish to make such claims then this should be explained in the intro/discussion citing the relevant studies and explaining the improvements compared to previous findings. For example, Soldner et al, 2016; Nature have used CRISPR/Cas9 genome editing to introduce a pathogenic variant in the distal enhancer of the SNCA gene thereby altering SNCA gene expression. Furthermore, in the same study the authors demonstrated that these sequence changes resulted in altered binding of EMX2 and NKX6-1 TFs that when over-expressed resulted in SNCA down-regulation. I understand that the binding was assessed by EMSA and that no CRISPR-guided TF reintroduction was done in the Soldner et al study, however in my opinion this does not constitute enough advance for such claims.

In the introduction, the part on lack of coherence between the ideas being pursued by those studying functional variants and those performing epigenetic editing is somewhat misleading. This is a very complex topic and it is likely that in some cases DNA methylation will block or facilitate TF binding (Domcke et al 2015, Nature; Pflueger et al, 2018, Genome Res) whereas in some other cases it will merely follow TF binding (Stadler et al, 2011, Nature; Thurman et al 2012 Nature). This will depend on the TF, cell line, organism etc, and there are certainly studies that support either scenario. This is explained somewhat better in the discussion, however, having this section placed in the introduction hints that the authors might be trying to address this point in the current manuscript. I also don't understand why the authors use "epigenetic editing", or a "form of epigenetic editing" to refer to the Cas9-mediated introduction of p65. This is especially strange given that the corresponding author recently published a guide to the usage of the word "epigenetics" where he specifically suggested the term "epigenetic" not to be used in such cases.

In the Results section the authors again make priority claims that are not justified. For example: "We also performed exploratory whole genome bisulphite sequencing (WGBS) assays pioneering the Illumina HiSeq X technology to profile DNA methylation throughout the genomes of these cell lines". WGBS libraries have previously been sequenced on the Illumina HiSeq X platforms. Please see: Nair et al, 2018: "Guidelines for whole genome bisulphite sequencing of intact and FFPE DNA on the Illumina HiSeq X Ten." (Epigenetics and Chromatin).

I am not sure if I understand the following filtering step (associated with Figure 1): "As a further filter, with the assumption that one allele being active and the other silenced should be associated with a pattern of intermediate DNA methylation locally, we chose the subset of loci in which CG dinucleotides with DNA methylation values of 20-80% were found within the peak." Average DNA methylation levels can reveal very little about allelic activity unless the authors have looked at the distributions of DNA methylation within sequenced reads. For example, a region that is 50% methylated can either be partially methylated, i.e. all reads are 50% methylated (such as in LMRs, for example) or 50% of the reads can be fully methylated and the other 50% fully unmethylated.

Figure 2a. This figure by itself is not very informative. It would be useful to overlay this with the wealth of published ENCODE data to demonstrate what kind of chromatin make-up is associated with this locus (DNAm, H3K4me1, p300, H3K27ac).

Figure 3 (a,d). What is on the y axis? Read numbers, FPKM or something else? Figure legend says "read pile-ups"? Does this mean that AA/HSA is covered by 4 reads only? This needs to be clarified. Also, I could not find a table / sup. Figure with the description of the datasets (sequencing depth, read nr, conversion % for BS-seq etc).

Figure 3 (c). What is "normalised gene counts"? (FPKM, RPKM, TPM?).

What is the coverage of the mCs / Cs in Figure 3e.

Representations such as the ones in Figure 3a, d, e can be very misleading if no surrounding regions are shown. The authors need to show ATAC/ChIP/DNAm signal throughout the locus and then the panels (a,d,e) as insets. Otherwise it is very difficult to judge the strength of the signal. I have similar concerns for Figure 4.

I would suggest the authors to perform CHIP-seq on the targeted p65 sample and adequate controls and to properly assess the specificity of the experiment presented in Figure 5.

Reviewer #3 (Remarks to the Author):

Johnston et al. seek to answer how functional variants modulate transcription using genome engineering and epigenetic editing. This paper is written in a somewhat philosophical tone that extrapolates the findings from one locus to explain an entire mode of genome regulation. To my knowledge this exact experiment of mutating a functional TF site and then targeting the TF to that locus to restore function has not been published and so this is a novel way of demonstrating this principle. The novelty of the paper rests on the combined CRISPR editing and dCas9-p65 approach and observation.

Main points:

To establish some degree of generality it would be great to have a second gene / TF site edited in this manner but if there is no other NFkB /p65 site this would require creation of a new dCas9 system which is likely beyond the scope of this paper. Is there another NFkB functional SNP site in the data?

In 3D the authors claim p65 binds to the AA allele in a cell line that has both an AA and HSA allele. If the authors can not distinguish between which allele contributes to the CHIP-seq signal then they should perform CHIP-seq for p65 in cells with only the AA allele and only the HAS allele to demonstrate the this SNP AA/HAS site is responsible for the NFkB binding at this genomic locus.

Specific points:

The paper is written in a strange and meandering way to make the main point on TF binding site vs TF recruitment. For example why are novel microRNAs discussed? There is a very large amount of text describing the admittedly impressive pipeline leading to the CRISPR experiments but if the main novelty is in the CRISPR TF/ dCas9-p65 experiment then why not move most of this text to the methods?

Line 65 the authors state that the premise of epigenetic editing is that TF may not be required. This is a somewhat strange interpretation of epigenetic editing papers that epigenetic edits likely act through recruitment of proteins which could include transcription factors to a locus. I suppose for unperturbed transcription this is a bit of a chicken or the egg question.

If the authors think their paper which uses RNA-seq and the newer haplotype assignments is superior then why do they compare to Li et al to trim their list? The degree of overlap is quite low and the authors should state why and which should be trusted.

I find the analysis of human specific transcription fascinating. In my opinion expanding the explanation of this in the results (eg primates vs humans vs Neanderthals) and discussion would add interest.

Thanks to the reviewers – their detailed and insightful comments prompted us to make a number of changes to the manuscript, as described below, resulting in a substantially improved report. We show where the manuscript has been edited using blue text in this revised submission.

Reviewer #1 (Remarks to the Author):

Johnson et al tried to answer the role of genomics variants on the TF factor binding site for the gene expression, chromatin accessibility and nearby DNA methylation status. There are two main experiments in this project. First, they aim to identify the polymorphic loci with very strong functional effects on gene expression, chromatin accessibility and DNA methylation. They used lymphoblastoid cell lines from one of the well-established pedigrees called Utah1463. In this part, they performed tremendous job to identify the locus in which 3 SNPs were found to control the TBC1D4 gene expression. Next, they modified those variants by using CRISPR genetic/epigenetic tools to support their hypotheses that TFs binding mediates the transcriptional regulation. If they answer some concerns in the second part, I think this paper will help us to understand the role of TFs and epigenetic (chromatin) changes and their crosstalk for the gene expression.

Major Point:

1. In Figure 3 and Figure 4, they demonstrated the loss of NFκB interaction with their target loci by using sequence-based experiment. To show the direct interaction between the TFs and its DNA-motif, they can perform EMSA staining by using anti-P65 or anti-P50 antibody. Super-shift in the EMSA will prove the physical interaction of the protein-DNA on their identified locus.

We didn't describe the evidence to support this in the original submission, but now we add a citation from Chen and colleagues (1998) – this is a very well-studied NFκB binding motif, there have been multiple studies over decades performing the kinds of *in vitro* assays described by the reviewer, including EMSAs. We add the Chen reference (reference 52, line 174) to point the reader to an example of this prior literature to address the reviewer's concern.

2. By performing ATAC-seq, they showed very little chromatin accessibility in the HSA/HSA individual with respect to the AA/AA. First, it will be better to demonstrate the peak on a large scale (100 bp on the both sides) in a supplemental figure. Then, they can look at the peak on the variant site for the well-known epigenetic markers. If H3K27ac mark is deposited in their locus, they can test the role of TF and epigenetic mark for the gene expression in their well-suited model. This will help scientific community to understand the current discussion between two groups working on functional variants and performing epigenetic editing. Recently, Hilton et al (Nat biotech, 2015) established dCas9-P300 system to deposit H3K27 mark on promoter and enhancer sites and Kuscu et al (JMB, 2018) used this system to make denovo mark on non-regulatory sites. If authors of this paper accumulate this histone mark on their target site in HSA/HSA cell line where NFκB cannot bind, they have a chance to see whether TF binding is necessary for the gene induction or not. If this is applicable, p300 will give more direct relationship for the role of epigenetic editing for gene expression than P65.

There is a lot in this comment. We have now updated multiple figures to show a broader genomic context around the multifunctional variant (the ~6 Mb haplotype overview in **Figure 2**, eQTL data from nearby genes in **Figure 4**, ATAC-seq and H3K27ac ChIP-seq for the same region in **Figure 5**, ATAC-seq in the 5 kb flanking the multifunctional variant in **Figure 6**, ChIP-seq for H3K27ac and NFκB in the same 5 kb region in **Figure 7** and DNA methylation in **Figure 8**). We agree that the broader context is useful in understanding what's happening at the site of the multi-functional variant itself.

The reviewer's suggestion that we should test whether the CRISPR targeting experiments cause a change in H3K27ac is exactly where we want to take this system in follow up experiments. The problem is that in the transient transfection approach we use at present, we do not get enough cells for the necessary ChIP experiments. We will have to re-derive the system with stably incorporated constructs that we can induce and look for a temporal response. We agree that this will be a great way of understanding the hierarchical cascade of events occurring in establishing a *cis*-regulatory element, and will be a focus of new experiments using this system.

3.They found their variant site 182kb away from the promoter of TBC1D4, and it seems that they said these variants can only alter the expression of the gene where the variants exist. In figure 4 and 5, they focus only on the TBC1D4 gene expression as well. It is reasonable to focus on the gene identified by QTL analysis; however, they need to show the effect of their genetic/epigenetic alterations on nearby gene to validate the specificity of the TFs for the transcription regulation of their eQTL Gene.

This is a great idea, and prompted us to include the data in **Figure 4**, which show that while *TBC1D4* is a target of the eQTL at the multifunctional variant, the nearby *COMMD6* and *UCHL3* genes within the larger interaction domain shown in **Figure 2** are not.

Minor Points:

Line 100-101: Two slow-grow cell lines(GM77 and GM93) were called as outliers for gene expression, however Supp Fig 2 did not show them as an outlier. In this supp figure, GM79 and GM80 were depicted as an outlier in addition to the GM93.

We decided to be completely transparent in our quality assessment studies, which leaves us open to legitimate questions like these. We were able to track down one other feature of GM(128)93, the unusual EBV expression pattern shown in **Figure S3** to explain its slow growth. Additionally, cell lines GM(128)79 and GM(128)80 were outliers for gene expression due to their separate batch for RNA extraction, which we have made more clearly in the text and **Figure S2**. We believe it is the better choice to acknowledge the variability and show it to the reader. As we describe, we only perform QTL analyses using the 11 children (cell line GM(128)77 is the father) and employ linear regression to remove sources of variability (Batch and EBV activity), so the variability should not persist to cause problems with interpretation of our results.

Line 179: Does ATAC-Seq data in Fig3a belong to only GM78 cell line or multiple cell lines? In figure legends, you mentioned that pileup data have been shown for AA/AA(n:2), AA/HSA(n:7) and HSA/HAS(n:2). GM78 has also not been recognized in the panel b and c of the Figure 3 as an AA/HSA sample.

The ATAC-seq data in the old Figure 3a (now **Figure 6a**) are indeed from different cell lines, now named in the revised figure. Throughout **Figures 5-7** we now show which cell line was the source of data shown.

Line 221: I think "HAS" should be "HSA".

That is correct, now fixed, thanks.

Fig2: There is inconsistency for the genomics positions in the panel A and B. If the red arrow shows the variant, then these variants in panel B should be around 75,300,000. Please check which version of the human genome (hg) was used for each panel.

We now show which version of the human reference genome is used in each figure, which helps to resolve the confusion when we had to use hg19 in **Figure 3**.

Fig3a: zoom-out image of the ATAC-seq profile. (for 100-200 bp around the variant region). It will give us a better idea for the position of the variants on or near the peak.

As described above, we now include the ATAC-seq data representation in the 5 kb flanking the multifunctional variant in **Figure 6**.

Fig4c: typo; change the "accessibility" into "accessibility".

Fixed, thanks.

FigS3: What is the pinkish bar on the top of the panel. What does color density represent?

We have now added the key for that heat map – quartiles of EBV expression.

FigS5: What is the size of the bands in the marker?

Sizes now added.

Reviewer #2 (Remarks to the Author):

The manuscript by Johnston and colleagues describes a conceptually simple experiment that demonstrates how the alteration of DNA sequence (a multi-functional variant) corresponding to NFκB binding sites alters NFκB binding and the expression of the TBC1D4 gene. The authors employed high-precision genome editing tools to test this in the LCL cell line and have generated a whopping 85 genomics data sets to support this hypothesis. Such studies are much needed as they can greatly aid in clarifying the function of diverse genomic variants. Nevertheless, it is fair to say that none of the reported findings come across as particularly surprising. Also, the manuscript makes a number priority claims that are not properly backed up by scientific data and are in my opinion unnecessary (see points 1 and 3). I believe that it would be much more useful to place this work in the context of already published literature and highlight the methodological improvements. The figures also need significant improvements as in many case it is not clear what is being presented and axis labels are not clearly explained. My suggestions can be found below.

The claim in the abstract: "In this study, we provide the first formal proof..." should be removed. If the authors wish to make such claims then this should be explained in the intro/discussion citing the relevant studies and explaining the improvements compared to previous findings. For example, Soldner et al, 2016; Nature have used CRISPR/Cas9 genome editing to introduce a pathogenic variant in the distal enhancer of the SNCA gene thereby altering SNCA gene expression. Furthermore, in the same study the authors demonstrated that these sequence changes resulted in altered binding of EMX2 and NKX6-1 TFs that when over-expressed resulted in SNCA down-regulation. I understand that the binding was assessed by EMSA and that no CRISPR-guided TF reintroduction was done in the Soldner et al study, however in my opinion this does not constitute enough advance for such claims.

We see where these concerns are coming from, and have no problem making the language a bit less enthusiastic (now "In this study, we provide direct proof to support this model"). As the reviewer notes, the *SNCA* example and other studies (including those cited from the Lappalainen group) show the strong indirect evidence for TFs mediating the effect of sequence variation, but without the restoration experiment we have performed. This restoration approach is new, and while the results are indeed unsurprising, it is reasonable to point out that we are showing something not previously demonstrated.

In the introduction, the part on lack of coherence between the ideas being pursued by those studying functional variants and those performing epigenetic editing is somewhat misleading. This is a very complex topic and it is likely that in some cases DNA methylation will block or facilitate TF binding (Domcke et al 2015, Nature; Pflueger et al, 2018, Genome Res) whereas in some other cases it will merely follow TF binding (Stadler et al, 2011, Nature; Thurman et al 2012 Nature). This will depend on the TF, cell line, organism etc, and there are certainly studies that support either scenario. This is explained somewhat better in the discussion, however, having this section placed in the introduction hints that the authors might be trying to address this point in the current manuscript. I also don't understand why the authors use "epigenetic editing", or a "form of epigenetic editing" to refer to the Cas9-mediated introduction of p65. This is especially strange given that the corresponding author recently published a guide to the

usage of the word “epigenetics” where he specifically suggested the term “epigenetic” not to be used in such cases.

There are a couple of separate issues here. To take the second issue first, our guide to the use of the word ‘epigenetic’ was not as prescriptive as the reviewer suggests; we instead say that when someone uses the word, they should go on to explain what they mean in that situation. We are doing so here: we use the term ‘epigenetic editing’ in quotation marks to try to convey the ambiguity of the use of the term, and go on to clarify that “‘epigenetic’ regulators are those influencing chromatin states and DNA methylation”, so that the reasonably commonly-used term (epigenetic editing) is explained.

We completely agree that the relationship between TF binding and other chromatin and DNA methylation regulatory mechanisms is a very complex topic and can’t really be discussed in sufficient detail in an introduction to a manuscript (and has prompted us to think of writing a review on this topic, thanks to the reviewer for the idea). The finding that *cis*-regulatory elements (where TFs bind) as a group are so strikingly unmethylated across the genome does not preclude a subset of TFs binding preferentially to methylated sites, and there is excellent *in vitro* evidence for some TFs having such a preference. However, we believe it is worthwhile making the point that opening up chromatin at a locus or removing DNA methylation (‘epigenetic editing’) may not by itself be as fruitful an approach as targeting a TF that can recruit a more complex set of mediators locally. The goal of that part of the Introduction is to get those performing ‘epigenetic editing’ to think more critically about what might be an effective approach in targeting transcriptional regulatory changes.

In the Results section the authors again make priority claims that are not justified. For example: “We also performed exploratory whole genome bisulphite sequencing (WGBS) assays pioneering the Illumina HiSeq X technology to profile DNA methylation throughout the genomes of these cell lines”. WGBS libraries have previously been sequenced on the Illumina HiSeq X platforms. Please see: Nair et al, 2018: “Guidelines for whole genome bisulphite sequencing of intact and FFPE DNA on the Illumina HiSeq X Ten. “ (Epigenetics and Chromatin).

The Nair *et al.* paper was very good, and our language was probably a bit too over-enthusiastic, we accept the criticism. We now say “We also performed exploratory whole genome bisulphite sequencing (WGBS) assays using our optimized Illumina HiSeq X-based assay⁴⁴ to profile DNA methylation throughout the genomes of these cell lines.”

I am not sure if I understand the following filtering step (associated with Figure 1): “As a further filter, with the assumption that one allele being active and the other silenced should be associated with a pattern of intermediate DNA methylation locally, we chose the subset of loci in which CG dinucleotides with DNA methylation values of 20-80% were found within the peak.” Average DNA methylation levels can reveal very little about allelic activity unless the authors have looked at the distributions of DNA methylation within sequenced reads. For example, a region that is 50% methylated can either be partially methylated, i.e all reads are 50%

methyated (such as in LMRs, for example) or 50% of the reads can be fully methyated and the other 50% fully unmethyated.

We're not sure that we understand this concern fully, so we want to make it clear here what we did and why. We do not average *in cis* across multiple CGs within a region, we measure the C/(C+T) ratio at a specific cytosine across all bisulphite sequencing reads, and then find the mean for that specific site across the 2 HS/HS, 7 AA/HS and 2 AA/AA individuals. At a locus with 20-80% DNA methylation, there have to be some alleles with methyated and some unmethyated cytosines. This could be due to some cells being methyated on both alleles and others on neither (a cellular mosaicism) or one parental allele being methyated and the other not (an intracellular difference), or a combination of the two. It is more likely that a heterozygous locus functional on one allele and not on the other would have a DNA methylation reading of 50% than <20% or >80%, so we used this as another way of prioritising loci. Our new way of phrasing the approach is hopefully helpful: "We then further filtered by requiring the locus to have an intermediate DNA methylation value, consistent with only one allele being active..."

Figure 2a. This figure by itself is not very informative. It would be useful to overlay this with the wealth of published ENCODE data to demonstrate what kind of chromatin make-up is associated with this locus (DNAm, H3K4me1, p300, H3K27ac).

We agree, and have now added haplotype and chromatin conformation information (**Figure 2**), and the broader contextual depictions in **Figures 5-8**, with chromatin and DNA methylation information portrayed.

Figure 3 (a,d). What is on the y axis? Read numbers, FPKM or something else? Figure legend says "read pile-ups"? Does this mean that AA/I is covered by 4 reads only? This needs to be clarified. Also, I could not find a table / sup. Figure with the description of the datasets (sequencing depth, read nr, conversion % for BS-seq etc).

We now clarify what each axis represents in the figures and figure legends throughout the paper. We also add a new **Table S7** with the descriptions of the datasets generated.

Figure 3 (c). What is "normalised gene counts"? (FPKM, RPKM, TPM?).

It's complex, but now added to figure – the value is an estimate derived from the program Sleuth's downstream processing of Kallisto pseudo-alignment. While not corrected for gene length, we only compare values of gene expression between samples, not between genes in a sample, so gene length is not a factor in causing differences.

What is the coverage of the mCs / Cs in Figure 3e.

These data are now added in **Table S6**.

Representations such as the ones in Figure 3a, d, e can be very misleading if no surrounding regions are shown. The authors need to show ATAC/ChIP/DNAme signal throughout the locus and then the panels (a,d,e) as insets. Otherwise it is very difficult to judge the strength of the signal. I have similar concerns for Figure 4.

As described in the response to Reviewer 1, who shared the concern (point #2 above), we have now done that in multiple figures.

I would suggest the authors to perform ChIP-seq on the targeted p65 sample and adequate controls and to properly assess the specificity of the experiment presented in Figure 5.

As mentioned in the response to the same issue (#2) of Reviewer 1, this is indeed where this work needs to proceed, but will require our building a stable cell line with inducible constructs in order to be able to generate enough cells for ChIP. We are in full agreement with the reviewer, but this has to represent follow up work, as it would take a major investment of time to accomplish.

Reviewer #3 (Remarks to the Author):

Johnston et al. seek to answer how functional variants modulate transcription using genome engineering and epigenetic editing. This paper is written in a somewhat philosophical tone that extrapolates the findings from one locus to explain an entire mode of genome regulation. To my knowledge this exact experiment of mutating a functional TF site and then targeting the TF to that locus to restore function has not been published and so this is a novel way of demonstrating this principle. The novelty of the paper rests on the combined CRISPR editing and dCas9-p65 approach and observation.

Main points:

To establish some degree of generality it would be great to have a second gene / TF site edited in this manner but if there is no other NFκB /p65 site this would require creation of a new dCas9 system which is likely beyond the scope of this paper. Is there another NFκB functional SNP site in the data?

We appreciate the concern, and we looked to see if any of the prioritised candidate multifunctional variants from **Figure 1** were predicted to disrupt an NFκB binding site, but none of the others on the list are predicted to do so. This will be a valuable suggestion to pursue in follow up work.

In 3D the authors claim p65 binds to the AA allele in a cell line that has both an AA and HAS allele. If the authors can not distinguish between which allele contributes to the CHIP-seq signal then they should perform CHIP-seq for p65 in cells with only the AA allele and only the HAS allele to demonstrate the this SNP AA/HAS site is responsible for the NFκB binding at this genomic locus.

We now show ATAC-seq (**Figure 6b**) and ChIP-seq for NFκB components (p65, RelB, RelC, p50 and p52, **Figure 7b**) results in heterozygous AA/HS samples with visualisation of reads with the AA haplotype, showing that this AA haplotype is indeed preferentially or solely responsible for the reads generated.

Specific points:

The paper is written in a strange and meandering way to make the main point on TF binding site vs TF recruitment. For example why are novel microRNAs discussed? There is a very large amount of text describing the admittedly impressive pipeline leading to the CRISPR experiments but if the main novelty is in the CRISPR TF/ dCas9-p65 experiment then why not move most of this text to the methods?

The miRNA studies are included in the first part of the study in which we wanted to identify very high confidence multifunctional variants. We now clarify the justification for this component of the study with the new test “We were concerned that a functional variant might be acting on the expression of a microRNA, which could then be acting on a gene to alter its mRNA levels, leading to the impression that the functional variant was acting on the gene directly. We therefore performed microRNA sequencing...”

Line 65 the authors state that the premise of epigenetic editing is that TF may not be required. This is a somewhat strange interpretation of epigenetic editing papers that epigenetic edits likely act through recruitment of proteins which could include transcription factors to a locus. I suppose for unperturbed transcription this is a bit of a chicken or the egg question.

It is indeed a chicken and egg question, as we highlight in the introduction. We responded to a similar question from Reviewer 2 above, in which we discuss the complexity of this issue.

If the authors think their paper which uses RNA-seq and the newer haplotype assignments is superior then why do they compare to Li et al to trim their list? The degree of overlap is quite low and the authors should state why and which should be trusted.

Our goal in the comparison with the Li et al. paper was to help identify high-confidence multifunctional variants, not to be comprehensive. Trimming the list to identify a very small number of candidates including one that would justify editing and re-targeting was what we needed. The Li *et al.* paper was attempting to be comprehensive, the two papers are not really comparable. Therefore, while we had the advantage of updated haplotypic information and our directional RNA-seq as opposed to mRNA-seq using oligo(dT)-capture, we would not like to give any impression that Li and colleagues did anything untrustworthy, we had a very different goal in our analysis.

I find the analysis of human specific transcription fascinating. In my opinion expanding the explanation of this in the results (eg primates vs humans vs Neanderthals) and discussion would add interest.

We agree, this is the result that our population genetics colleagues are especially enthusiastic about. There's not much more to say about this finding than we are already presenting, but it would be very interesting to see if *TBC1D4* becomes of interest in a phenotypic association that has a difference in incidence in East Asian compared with European/African populations, for example.

REVIEWERS' COMMENTS:

Reviewer #1 (Remarks to the Author):

In this manuscript, Johnson et al focused on the role of functional variants near TF binding sites. The main claim of this paper is that functional variants around TF motifs can "directly" alter the TF binding that has an effect on gene expression, DNA Methylation, and chromatin states. This manuscript will help the scientific community to understand the crosstalk between the transcription factors and epigenetic changes as well. Their findings are novel and they used strong statistical analysis to support their hypothesis.

They responded to my three major concerns and minors as well.

Briefly,

for major-1: They answered it with extra reference from the literature.

for major-2: They updated their figures(fig 4,5,6,7 and 8) and focused on a broader genomic context around the multifunctional variant. Now, we have a better idea for the role of TFs on their neighboring loci as well. They are also optimizing their system for their follow-up experiment and they can answer the hierarchical cascade during transcription in their future studies. Their findings from this study will let them do this.

for major 3: They performed an extra experiment according to my suggestion, and included in the updated Figure 4-b. Their results satisfied my concern.

They responded to all minor concerns and fixed them in the revised version.

reviewed by Cem Kuscu

Reviewer #2 (Remarks to the Author):

The authors have answered all of my queries (and those of other reviewers). This work is one of the best examples of how alterations in the DNA sequence exercise downstream effects via altering TF binding.

Reviewer #3 (Remarks to the Author):

The authors have addressed all my concerns.